# A convolutional neural-network framework for modelling auditory sensory cells and synapses

Fotios Drakopoulos [1✉], Deepak Baby [1] & Sarah Verhulst [1]

In classical computational neuroscience, analytical model descriptions are derived from neuronal recordings to mimic the underlying biological system. These neuronal models are typically slow to compute and cannot be integrated within large-scale neuronal simulation frameworks. We present a hybrid, machine-learning and computational-neuroscience approach that transforms analytical models of sensory neurons and synapses into deep-neural-network (DNN) neuronal units with the same biophysical properties. Our DNN-model architecture comprises parallel and differentiable equations that can be used for back-propagation in neuro-engineering applications, and offers a simulation run-time improvement factor of 70 and 280 on CPU or GPU systems respectively. We focussed our development on auditory neurons and synapses, and show that our DNN-model architecture can be extended to a variety of existing analytical models. We describe how our approach for auditory models can be applied to other neuron and synapse types to help accelerate the development of large-scale brain networks and DNN-based treatments of the pathological system.

[1] Department of Information Technology, Ghent University, Ghent, Belgium. ✉email: fotios.drakopoulos@ugent.be

Following the fundamental work of Hodgkin and Huxley in modelling action-potential generation and propagation[1], numerous specific neuronal models were developed that proved essential for shaping and driving modern-day neuroscience[2]. In classical computational neuroscience, transfer functions between stimulation and recorded neural activity are derived and approximated analytically. This approach resulted in a variety of stimulus-driven models of neuronal firing and was successful in describing the non-linear and adaptation properties of sensory systems[3–6]. For example, the mechano-electrical transduction of cochlear inner-hair-cells (IHCs) was described using conductance models[7–10] and the IHC-synapse firing rate using multi-compartment diffusion models[11–13]. Such mechanistic models have substantially improved our understanding of how individual neurons function, but even the most basic models use coupled sets of ordinary differential equations (ODEs) in their descriptions. This computational complexity hinders their further development to simulate more complex behaviour, limits their integration within large-scale neuronal simulation platforms[14,15], and their uptake in neuro-engineering applications that require real-time, closed-loop neuron model units[16,17].

To meet this demand, neuroscience recently embraced deep learning[18], a technique that quickly revolutionised our ability to construct large-scale neuronal networks and to quantify complex neuronal behaviour[19–27]. These machine-learning methods can yield efficient, end-to-end descriptions of neuronal transfer functions, population responses or neuro-imaging data without having to rely on detailed analytical descriptions of the individual neurons responsible for this behaviour. Deep neural networks (DNNs) learn to map input to output representations and are composed of multiple layers with simplified units that loosely mimic the integration and activation properties of real neurons[28]. Examples include DNN-based models that were successfully trained to mimic the representational transformations of sensory input[29,30], or DNNs that use neural activity to manipulate sensory stimuli[31,32]. Even though deep learning has become a powerful research tool to help interpret the ever-growing pool of neuroscience and neuroimaging recordings[33,34], these models have an important drawback when it comes to predicting responses to novel inputs. DNNs suffer from their data-driven nature that requires a vast amount of data to accurately describe an unknown system, and can essentially be only as good as the data that were used for training. Insufficient experimental data can easily lead to overfitted models that describe the biophysical systems poorly while following artifacts or noise present in the recordings[35]. The boundaries of experimental neuroscience and associated limited experiment duration hence pose a serious constraint on the ultimate success of DNN-based models of neuronal systems.

To overcome these difficulties and merge the advantages of analytical and DNN model descriptions, we propose a hybrid approach in which analytical neuronal models are used to generate a sufficiently large and diverse dataset to train DNN-based models of sensory cells and synapses. Combinations of traditional and machine-learning approaches were recently adopted to optimise analytical model descriptions[36–38], but our method moves in the opposite direction and takes advantage of deep-learning benefits to develop convolutional-neural-network (CNN) models from mechanistic descriptions of neurons and synapses. We show here that the resulting CNN models can accurately simulate outcomes of state-of-the-art auditory neuronal and synaptic diffusion models, but in a differentiable and computationally efficient manner. The CNN-based model architecture is compatible with GPU computing and facilitates the integration of our model units within large-scale, closed-loop, or spiking neuronal networks. The most promising design feature

relates to the backpropagation property, a mathematically complex trait to achieve for non-linear, coupled ODEs of traditional neural models. We will illustrate here how normal and pathological CNN models can be used in backpropagation to modify the sensory stimuli to yield an optimised (near-normal) response of the pathological system.

We develop and test our hybrid approach on sensory neurons and synapses within the auditory system. The cochlea, or inner-ear, encodes sound via the inner hair cells (IHCs). IHCs sense the vibration of the basilar membrane in response to sound using their stereocilia and translate this movement into receptor potential changes. By virtue of $Ca^{2+}$-driven exocytosis, glutamate is released to drive the synaptic transmission between the IHC and the innervated auditory-nerve fiber (ANF) synapses and neurons[39]. Experimentally extracted IHC parameters from in-vitro, whole-cell patch clamp measurements of the cellular structures and channel properties[40,41] have led to different model descriptions of the non-linear and frequency-dependent IHC transduction[10,42–44]. Parameters for analytical IHC–ANF synapse models are mainly derived from single-unit auditory-nerve (AN) recordings to basic auditory stimuli in cats and small rodents[45–51]. Progressive insight into the function of IHC–ANF synapses over the past decades has inspired numerous analytical model descriptions of the IHC, IHC–ANF synapse, and ANF neuron complex[11–13,52–62].

To generate sufficient training data for our CNN-based models of IHC–ANF processing, we adopted a state-of-the-art biophysical model of the human auditory periphery that simulates mechanical as well as neural processing of sound[61]. We describe here how the CNN model architecture and hyperparameters can be optimised for such neuron or synapse models and we evaluate the quality of our CNN models on the basis of key IHC–ANF complex properties described in experimental studies, i.e., IHC excitation patterns, AC/DC ratio[63] and potential-level growth[8,40], ANF firing rate, rate-level curves[48,64], and modulation synchrony[51,65]. The considered evaluation metrics had stimulation paradigms that were not part of the training dataset, to allow for a fair evaluation of the neuroscientific properties of the trained CNN models. These metrics stem from classical neuroscience experiments that characterised the presynaptic IHC receptor potential and postsynaptic AN processing, and together they form a critical evaluation of the adaptation, tuning, and level-dependent properties of the IHC–ANF complex. After determining the final CNN model architectures, we compute their run-time benefit over analytical models and investigate the extent to which our methodology is applicable to different existing mechanistic descriptions of the IHC–ANF complex. Lastly, we provide two use cases: one in which IHC–ANF models are connected to a CNN-based cochlear mechanics model (CoNNear$_{cochlea}$[66]) to capture the full transformation of acoustic stimuli into IHC receptor potentials and ANF firing rates along the cochlear tonotopy and hearing range, and a second one where we illustrate how backpropagation can be used to modify the CNN model input to restore a pathological output.

## Results

Figure 1a depicts the adopted training and evaluation method to calibrate the parameters of each CoNNear module. Three modules that correspond to different stages of the reference analytical auditory periphery model[61] were considered: cochlear processing, IHC transduction and ANF firing. The calibration of the cochlear mechanics module (CoNNear$_{cochlea}$) is described elsewhere[66,67]; here we focus on developing the sensory neuron models (i.e., CoNNear$_{IHC}$ and CoNNear$_{ANF}$). Figure 1a illustrates the training procedure for the CoNNear$_{ANF_L}$ module of a low-spontaneous-

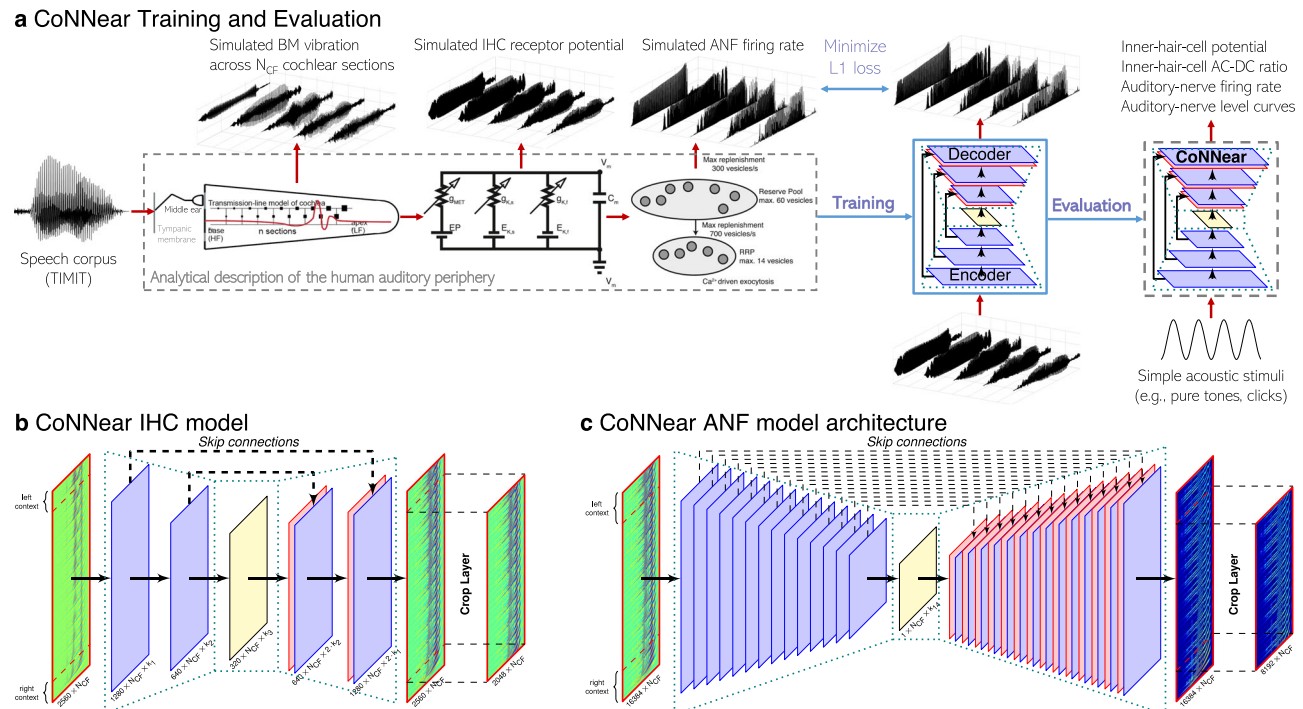

**Fig. 1 CoNNear overview. a** Overview of the CoNNear model training and evaluation procedure. **b** Architecture of the CoNNear inner-hair-cell transduction model. **c** Generic architecture used for the CoNNear auditory-nerve-fiber synapse models.

rate ANF. Acoustic speech material is given as input to the analytical descriptions of cochlear and IHC–ANF processing, after which simulated ANF firing rates are used as training material to determine the CoNNear$_{\text{ANF}_L}$ parameters. CoNNear modules were trained separately for each stage of the IHC–ANF complex, resulting in one model for IHC transduction and three models for different ANF types: a high- (H; 68.5 spikes/s), medium- (M; 10 spikes/s), and low- (L; 1 spike/s) spontaneous-rate (SR) ANF. We chose a modular approach because this facilitates future simulations of the pathological system, where the IHC receptor potential can be impaired through presbycusis[68], or where selective damage to the ANF population can be introduced through cochlear synaptopathy[69].

Each module was modelled using a convolutional encoder–decoder architecture, consisting of a distinct number of CNN layers, as shown in Fig. 1b, c. Within these architectures, each CNN layer is comprised of a set of filterbanks followed by a non-linear operation[18], except for the last layer where the non-linear operation was omitted. These end-to-end architectures process input waveforms of length $L_c$ across $N_{\text{CF}}$ frequency channels to generate outputs of the same size $L_c \times N_{\text{CF}}$. The encoder CNN layers use strided one-dimensional convolutions, i.e., the filters are shifted by a time-step of two to halve the temporal dimension after every CNN layer. Thus, after $N$ encoder CNN layers, the input signal is encoded into a representation of size $L_c/2^N \times k_N$, where $k_N$ equals the number of filters in the $N^{\text{th}}$ CNN layer. The decoder uses $N$ deconvolution, or transposed-convolutional, layers, to double the temporal dimension after every layer to re-obtain the original temporal dimension of the input ($L_c$). Skip connections were used to bypass temporal information from encoder to decoder layers to preserve the stimulus phase information across the architecture. Skip connections have earlier been adopted for speech enhancement applications to avoid the loss of temporal information through the encoder compression[70–73] and can benefit the model training to best simulate non-linear and level-dependent properties of

auditory processing by providing interconnections between several CNN layers[66,74]. Lastly, the input dimension $L_c$ included a number of previous and following input samples, to provide context information to the CoNNear modules when simulating an input of length $L$. Because CNN models treat each input independently, providing context is essential to avoid discontinuities at the simulation boundaries and take into account neural adaptation processes[66]. A final cropping layer was added to remove the context after the last CNN decoder layer.

To provide realistic input to the IHC–ANF models for training, acoustic speech waveforms were input to the cochlear model and the simulated cochlear basilar-membrane (BM) outputs were used to train and evaluate the IHC–ANF models. To this end, the IHC transduction model was trained using $N_{\text{CF}} = 201$ cochlear filter outputs with centre frequencies (CFs) that span the human hearing range (0.1–12 kHz) and that were spaced according to the Greenwood place-frequency description of the human cochlea[75]. Similarly, simulated IHC receptor potentials of the analytical model cochlear regions ($N_{\text{CF}} = 201$) were used as training material for the different ANF models. It should be noted that even though we trained the models on the basis of 201 inputs of fixed length $L$, the optimal weights for a single CF-independent IHC or ANF model were determined during the training phase. Thus, these model units can afterwards be connected to inputs of any length $L$ or $N_{\text{CF}}$ to simulate CF-dependent IHC or ANF processing of the entire cochlea.

To evaluate the CoNNear IHC–ANF models, it is important to characterise their properties to acoustic stimuli that were not seen during training. Training was performed using a single speech corpus[76], but IHC and ANF processing have very distinct adaptation, and frequency- and level-dependent properties to basic auditory stimuli such as tones, clicks, or noise. Hence, to test how well the CoNNear modules generalise to unseen stimuli and whether they capture key properties of biological IHC–ANF processing, we evaluated their performance on a set of classical experimental neuroscience stimuli and recordings that

**Table 1 Final parameter selection of the CoNNear architectures.**

| Parameters | $L$ | $L_l$ | $L_r$ | $L_c$ | Total layers | Filters/ layer | Filter length | Encoder activation | Decoder activation |
|---|---|---|---|---|---|---|---|---|---|
| CoNNear$_{IHC}$ | 2048 | 256 | 256 | 2560 | 6 | 128 | 16 | tanh | sigmoid |
| CoNNear$_{ANF_H}$ | 8192 | 7936 | 256 | 16384 | 28 | 64 | 8 | PReLU | PReLU |
| CoNNear$_{ANF_M}$ | 8192 | 7936 | 256 | 16384 | 28 | 64 | 8 | PReLU | PReLU |
| CoNNear$_{ANF_L}$ | 8192 | 7936 | 256 | 16384 | 28 | 64 | 8 | tanh | sigmoid |

The input length of each model was $L_c = L_l + L + L_r$, and the output length (after cropping) $L$ samples. The specified lengths $L$ were used during training, but each architecture can process inputs of variable lengths $L$ after training.

characterise IHC transduction and ANF firing. The six considered evaluation metrics (described in Methods) together form a thorough evaluation of the CoNNear IHC–ANF complex, and outcomes of these simulations were used to optimise the final model architecture and its hyperparameters. Lastly, to study the application range of our framework, we evaluated how well it generalises to other existing IHC–ANF model descriptions. Additional details on the model architecture, training procedure, and IHC–ANF evaluation metrics are given in Methods.

**Determining the CoNNear hyperparameters**. Table 1 shows the final layouts of all the CoNNear modules we obtained after taking into account: (i) the L1-loss on the training speech material (i.e., the absolute difference between simulated CNN and analytical responses), (ii) the desired auditory processing characteristics, and (iii) the computational load. The L1-loss was considered during training to determine the epochs needed to train each module and to get an initial indication of the architectures that best approximate the IHC–ANF model units. The auditory processing characteristics of each trained architecture were then evaluated on the basis of the six evaluation metrics to determine which architectures provide the most biophysically realistic description of the IHC–ANF complex. Our primary concern was to develop a biophysically realistic CNN model of the IHC–ANF complex, hence computational time was not the primary goal. However, where possible, we limited the hyperparameters of the architectures to keep the number of trained parameters (and associated computational complexity) as low as possible without compromising on the biophysical properties. Here, we describe the principled fine-tuning approach we followed for each CoNNear module architecture and additional details are given in Methods. For each CoNNear module, we first describe the initial set of hyperparameters that we kept fixed, and then motivate how the remaining hyperparameters were chosen to best predict the biophysical response properties of the reference mechanistic models.

**CoNNear IHC model**
*Fixed parameters*. We opted for an architecture with 6 convolution layers and a filter length of 16 to capture the computations performed by the analytical IHC model[61]. In each layer, 128 convolution filters were used and the input length was set to $L_c = 2048 + 2 \times 256 = 2560$ samples (102.8 ms). The initial architecture was based on an existing CNN model we fine-tuned for cochlear processing[66], which we adjusted based on the shorter adaptation time constants associated with IHC processing[63]. Specifically, we decreased the number of layers from 8 to 6 and the filter lengths from 64 to 16. Since the level- and frequency-dependent non-linear characteristics of IHC processing are not expected to differ much from cochlear processing, the same number of convolution filters was used in each layer (128). We evaluated a number of architectures with different layer numbers, filter numbers or filter durations, but these did not show

significant L1-loss improvements on the training set over the chosen architecture. The hyperparameter selection procedure is extended in Discussion, where we explain how the relationship between the adaptation properties of mechanistic models and the selected CNN architecture can be empirically quantified.

*Optimised hyperparameters*. The shape of the activation function, or non-linearity, is crucial to enable CoNNear to learn the level-dependent cochlear compressive growth properties and negative signal deflections present in BM and IHC processing. A *tanh* non-linearity was initially preferred for each CNN layer, since it shows a compressive characteristic similar to the outer-hair-cell (OHC) and IHC input/output function[8,77] and crosses the *x*-axis. Figure 2 shows that the trained architecture (b) generally followed the pure-tone excitation patterns (Metric 1) of the reference model (a), but showed a rather noisy response across CF, especially for the higher stimulation levels. To optimise the trained IHC model, different non-linear activation functions were compared for the encoder and decoder layers. Because the IHC receptor potential is expressed as a (negative) voltage difference, we opted for a *sigmoid* non-linear function in the decoding layers to better capture the reference model outputs, while ensuring that the compressive nature present in the *tanh* could be preserved. Figure 2c shows that using a *sigmoid* activation function instead of a *tanh* for the decoder layers outperformed the *tanh* architecture (b) and better predicted the excitation patterns of the reference model (a). Our selection is further supported by Supplementary Fig. 1, that shows the root-mean-square error (RMSE) between the simulated reference and CoNNear IHC model excitation patterns for six different stimulus frequencies.

Figure 3 furthermore depicts how the different activation-function combinations affected the simulated AC/DC ratios of the IHC responses across CF (Metric 2), and the half-wave rectified IHC receptor potential as a function of stimulus level (Metric 3). The logarithmic decrease of the AC/DC ratio and the linear-like growth of the IHC potential were predicted similarly using both architectures, but the *tanh* architecture overestimated the responses for high stimulus frequencies and levels. Overall, a much smoother response was achieved when using a *sigmoid* activation function in the decoder layers, motivating our final choice for the CoNNear IHC architecture (Table 1).

**CoNNear ANF models**
*Fixed parameters*. Our modular approach enables the use of preceding CoNNear stages to optimise our ANF model parameters. To determine a suitable architecture, we first took into account the longer adaptation time constants (and the associated slow decay to the steady-state response) of the analytical ANF model description compared to the adaptation time constants accociated with cochlear or IHC processing[12]. Figure 4a visualises the exponential decay over time of simulated ANF firing rates of the reference mechanistic model to sustained stimulation, i.e., for

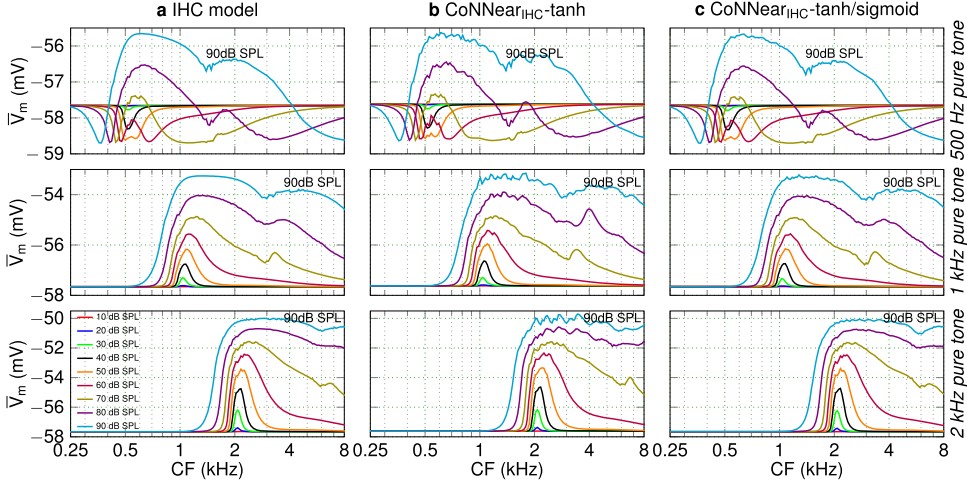

**Fig. 2 Comparing IHC excitation patterns (Metric 1).** Simulated average IHC receptor potentials $\overline{V}_m$ across CF for tone stimuli, presented at levels between 10 and 90 dB SPL. From top to bottom, the stimulus tone frequencies were 500 Hz, 1 kHz, and 2 kHz, respectively.

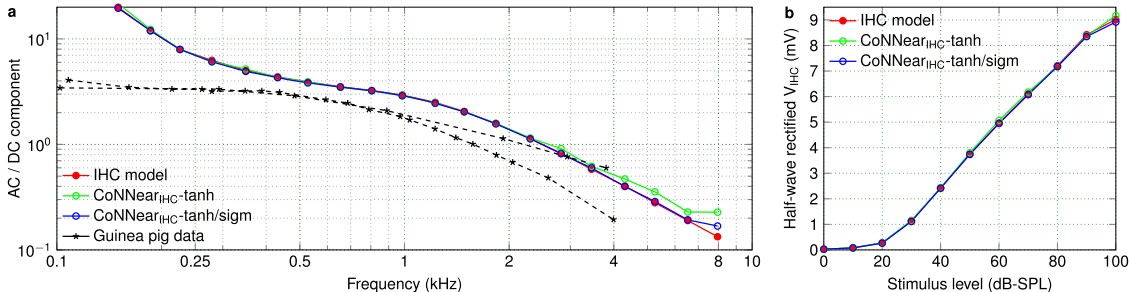

**Fig. 3 Comparing IHC transduction aspects. a** Ratio of the AC and DC components of the IHC responses (Metric 2) across CF for 80-dB-SPL pure-tone bursts compared against guinea-pig data[63]. **b** Root-mean-square of the half-wave rectified IHC receptor potential $V_{IHC}$ in response to a 4-kHz pure tone plotted as function of sound level (Metric 3).

an acoustic pure tone. Since CNNs treat each input window independently, the choice of the window size $L$ is important as it will determine the time dependencies that our ANF models will be able to encode and capture after training. Figure 4a shows that at the time corresponding to a window size of 2048 samples (~100 ms for $f_s = 20$ kHz), the firing rates of the three ANFs have not significantly decayed to their steady state and hence we opted for a longer window duration $L$ of 8192 samples ( ~400 ms). At 400 ms, the firing rates of the HSR, MSR, and LSR fibers have, respectively, reached 99.5, 95, and 93.4 % of their final (1-second) firing rate (Fig. 4a), providing a realistic description of the ANF adaptation properties to the CNN architecture.

Another important factor in the architecture design relates to capturing the experimentally[50] and computationally[61] observed slow recovery of the ANF onset-peak response after prior stimulation. The duration of the context window preceding the input window will be crucial to sufficiently capture the effect of prior stimulation on the response of CoNNear ANF models. Figure 4b shows the exponential recovery of the onset peak, for simulated responses of the three ANF types, as a function of the interstimulus interval between a pair of pure tones. Since the longest (1.9-second) interval corresponds to 38,000 samples, we compromised to select a final context window that was short enough to limit the computational complexity of the architecture, while still being able to capture the recovery properties of the reference ANF models faithfully. We chose 7936 samples for the left context window (~400 ms) which resulted in a total input size of $L_c = 7936 + 8192 + 256 = 16384$ samples. For a 400-ms interstimulus interval, the onset peak of the HSR, MSR, and

LSR fibers has recovered to the 92.4, 94.2, and 95.8 % of the onset peak of the 1.9-s interval tone, respectively (Fig. 4b). Additional information regarding the context window selection is provided in Methods and Supplementary Fig. 2.

Even though the selected window size $L$ of the CNN architecture adequately captures the ANF response time course, the length of the input that the architecture can actually encode is determined by the number of strided convolutional layers and the length of the filters in each layer. Thus, a much deeper architecture than chosen for the IHC model is required to capture the slower adaptation properties of the ANF responses to step-like stimuli (Fig. 4a). As shown in Fig. 4c, a trained architecture with 16 layers still failed to capture the exponential decay of the LSR ANF response, i.e., the fiber type with the slowest adaptation properties. By further increasing the receptive field of the CNN architecture (Eq. (4)) and encoding the input to a maximally condensed time representation, we can ensure that the long-term correlations existent in the input can be captured by the convolutional filters and the adaptation properties can be faithfully described by the resulting architecture. To this end, we opted for an architecture of 28 total layers and a filter length of 8, that downsamples the input size of $L_c = 16384$ samples to a condensed representation of size 1 in its encoder. The selected architecture was able to accurately capture the adaptation of the reference LSR ANF firing rate over time, as shown in the bottom panel of Fig. 4c. Lastly, we were able to decrease the number of filters in each layer from 128 (IHC) to 64 without compromising the L1-loss on the training dataset and the evaluated auditory characteristics of the trained architecture.

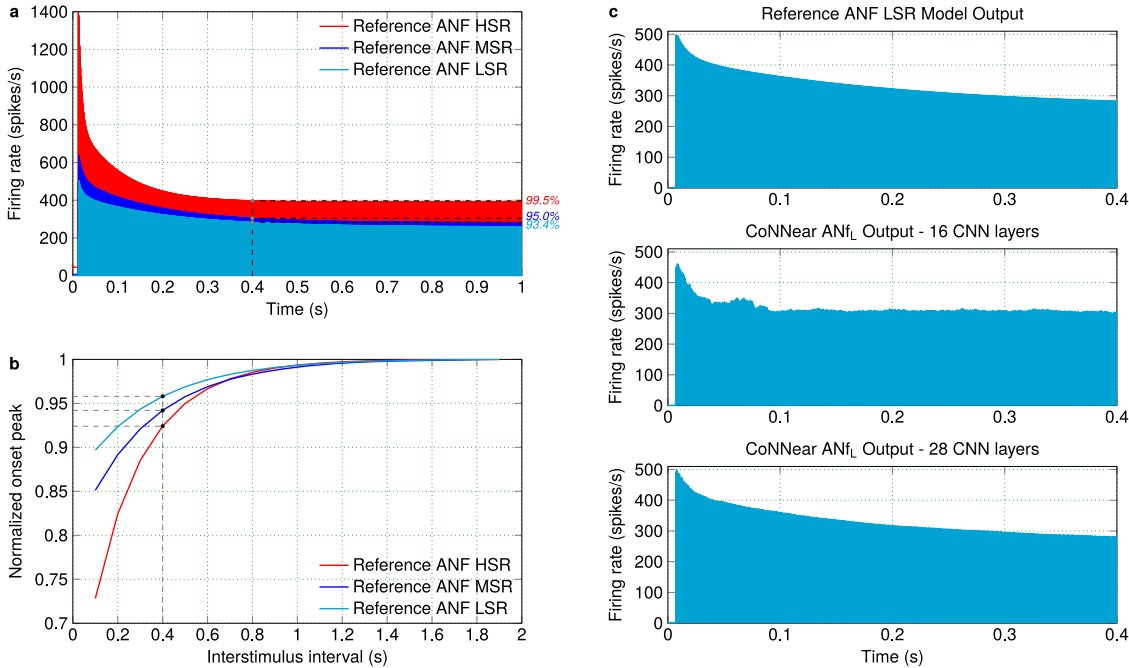

**Fig. 4 Parameter selection for the ANF models. a** The firing rate of the three ANF models is shown over time, as a response to a 1-second long 1-kHz pure tone presented at 70 dB SPL. The percentages on the right side correspond to the percent of the steady-state value that the firing rate reached at 0.4 s for each fiber. **b** The normalised amplitude of the onset peak is shown for a pair of 2-kHz pure tones with interstimulus intervals from 0.1 to 1.9 s. According to experimental procedures[50], 100-ms pure tones were presented at 40 dB above the firing threshold of each ANF (60, 65, and 75 dB for the HSR, MSR, and LSR fibers, respectively). Each time, the response maximum to the second tone is reported, normalised by the response maximum to the second tone with the longest interstimulus interval (1.9 s). **c** From top to bottom, the simulated ANF LSR firing rate to a 70-dB-SPL pure tone is shown for the reference ANF model, a trained model with 8 encoder layers and a trained model with 14 encoder layers.

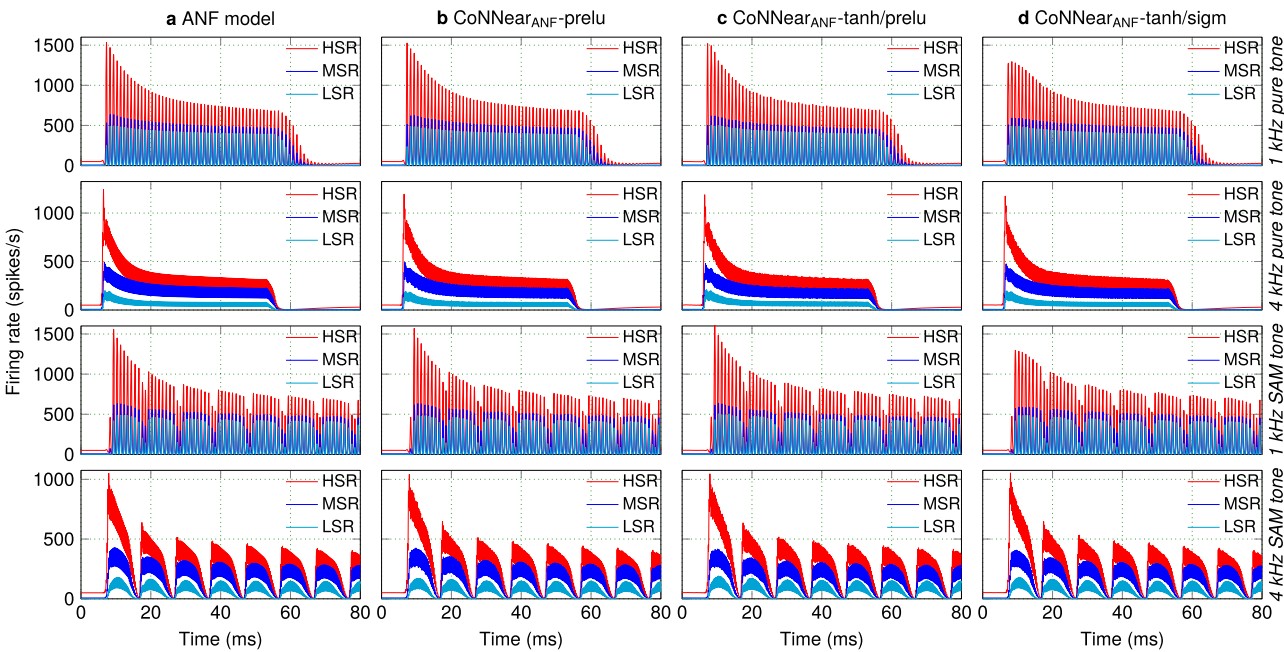

**Fig. 5 Comparing firing rates across different ANF models (Metric 4).** Simulated ANF firing rate across time for tone stimuli presented at 70 dB SPL. The blue, red, and cyan graphs correspond to the responses of the HSR, MSR, and LSR fiber models, respectively. From top to bottom, the stimuli were 1 kHz, 4 kHz pure tones and 1 kHz, 4 kHz amplitude-modulated tones.

*Optimised hyperparameters.* The compressive properties of BM and IHC processing are not observed in ANF processing, so a linear activation function (a Parametric ReLU; *PReLU*) was initially used for each CNN layer. Figure 5 shows the responses of the three trained CoNNear ANF models (b) for different tonal stimuli in comparison to the reference ANF model (a). The firing rates (Metric 4) of the three ANF models, CoNNear$_{ANF_H}$, CoNNear$_{ANF_M}$, and CoNNear$_{ANF_L}$, are visualised in red, blue, and cyan, respectively.

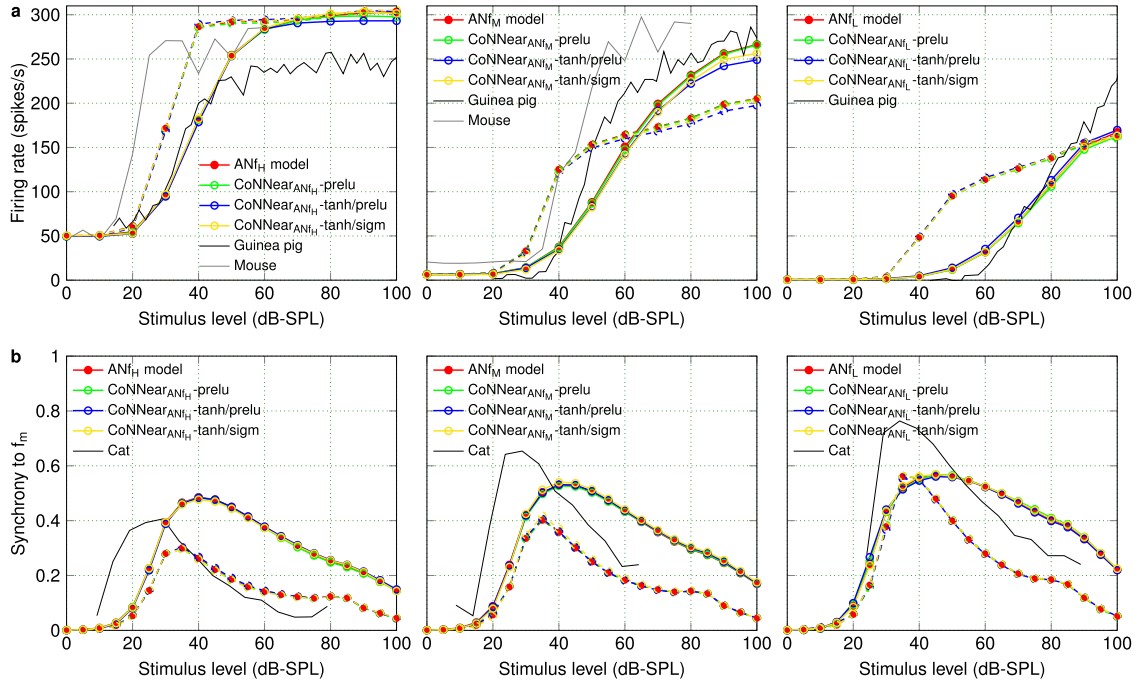

**Fig. 6 Level-dependent properties of the different ANF models. a** From left to right, ANF rate-level curves (Metric 5) were simulated for the HSR, MSR, and LSR ANF models, respectively, at CFs of 1 (dashed lines) and 4 kHz (solid lines). The reference data stemmed from guinea pig (fibers with SRs of 65, 10, and 0 spikes/s at a CF of ~1.5 kHz; Fig. 1 in ref. [48]) and mouse recordings (CF of 18.8 kHz for SR of 47.6 spikes/s and CF of 23.7 kHz for SR of 0.1 spikes/s; Fig. 6 in ref. [64]). **b** From left to right, ANF synchrony-level functions (Metric 6) were calculated for the HSR, MSR, and LSR ANF models. For each ANF model, 1 kHz and 4 kHz pure-tone carriers were modulated by an $f_m = 100$ Hz pure tone and presented at CFs of 1 (dashed) and 4 kHz (solid). For each CF, vector strength to the $f_m$ is reported against the stimulus intensity for the three fiber types. The reference data came from cat ANF recordings (fibers of 8.1 kHz CF and 2.6 spikes/s, 1.14 kHz CF and 6.3 spikes/s, and 2.83 kHz and 73 spikes/s, respectively; Figs. 5 and 8 in ref. [51]).

The good match between analytical and CoNNear predictions in Fig. 5 was extended to ANF rate- and synchrony-level growth as well (Fig. 6; Metrics 5 and 6), and together, these simulations show that the chosen architecture and *PReLU* non-linearity were suitable to model the three ANF types. Compared to the reference firing rates, the architectures in panel (b) introduced noise, that we attempted to eliminate by using a more compressive activation function (*tanh*) between the encoder layers. The *tanh* function was able to transform the negative potential of the IHC stage to the positive firing response of the ANFs (Fig. 5c) and yielded similar firing rates for all ANF models. However, for the CoNNear$_{ANF_H}$ and CoNNear$_{ANF_M}$ architectures, the *tanh* non-linearity introduced an undesired compressive behaviour at higher stimulus levels, as depicted in Fig. 6a. This was not the case for CoNNear$_{ANF_L}$, and hence we also tested whether using a *sigmoid* non-linearity in the decoder layers would further improve the predicted responses. Although this combination of non-linearities (d) compressed the responses of the CoNNear$_{ANF_H}$ and CoNNear$_{ANF_M}$ models even more, this combination was found to best approximate the firing rates of the analytical LSR ANF model. We further quantified the observed trends by calculating the RMSEs between the reference and CoNNear ANF firing rates to the same stimuli across different levels (Supplementary Fig. 3).

**Evaluating the biophysical properties of CoNNear IHC–ANF.** Even though different CNN architectures can be trained to yield a sufficiently low L1-loss on the speech dataset, it is important that the final CoNNear model also matches the biophysical properties of the mechanistic models and eletrophysiological recordings. To this end, we evaluated the performance of the CoNNear modules on six electrophysiology-based experiments using stimuli that were not part of the training set (Metrics 1–6). The excitation patterns of the final CoNNear IHC model (Fig. 2c) were generally consistent with the reference IHC model (a). The IHC AC/DC components (Fig. 3a) followed the simulated and measured curves well, and showed a slight overestimation for the lower frequency responses. The simulated half-wave rectified IHC receptor potential (Fig. 3a) corroborated the in-vivo guinea-pig IHC measurements[78] by showing an unsaturated, almost linear, growth of the half-wave rectified IHC receptor potential for stimulation levels up to 90 dB.

For each ANF model, the final CoNNear architectures (Table 1) followed the reference model firing rates across time (Fig. 5). As expected, phase-locking to the stimulus fine structure was present for the 1-kHz ANF response and absent for the 4-kHz ANF. Phase-locking differences between the 1 and 4-kHz CF fibers were also evident from their responses to amplitude-modulated tones. The level-dependent properties of different ANF types were also captured by our CoNNear architectures, as shown in Fig. 6. Compared to the reference data, the 4-kHz simulations captured the qualitative differences between LSR, MSR, and HSR guinea-pig ANF rates well. The mouse rate-level curves show somewhat steeper growth than our simulations, especially when comparing the lower SR fiber data with the simulated MSR fiber responses. Given that the cochlear mechanics are fundamentally different across species, it is expected that the level- and CF-dependence of the ANF responses are not overly generalisable across species. The shape of the simulated rate-level curves was different for the different CF fibers (1-kHz dashed lines compared to 4-kHz solid lines) despite the use of CF-independent parameters in the ANF model. This illustrates that differences in BM processing across CF, given as input to the IHC–ANF model, are still reflected in the shape of ANF rate-level curves. The smaller dynamic range of levels encoded by the BM for the 1-kHz than the 4-kHz CF (e.g.,

**Table 2 CoNNear execution time.**

| Model | Trainable parameters | Window (ms) | CPU (s) | | GPU (ms) | |
|---|---|---|---|---|---|---|
| | | | 201-CF | 1-CF | 201-CF | 1-CF |
| IHC model | - | 102.4 | 1.2707 | 0.6117 | - | |
| CoNNear$_{IHC}$ | 1,317,505 | 102.4 | 1.0262 | 0.0102 | 56.40 | 2.18 |
| ANF$_H$ model | - | 819.2 | 1.0553 | 0.7197 | - | |
| CoNNear$_{ANF_H}$ | 1,250,177 | 819.2 | 2.6792 | 0.0289 | 178.25 | 7.21 |
| ANF$_M$ model | - | 819.2 | 1.0508 | 0.7015 | - | |
| CoNNear$_{ANF_M}$ | 1,250,177 | 819.2 | 2.6820 | 0.0279 | 175.97 | 6.95 |
| ANF$_L$ model | - | 819.2 | 1.0590 | 0.7019 | - | |
| CoNNear$_{ANF_L}$ | 1,248,449 | 819.2 | 2.2074 | 0.0243 | 115.86 | 4.53 |
| IHC–ANF model | - | 819.2 | 9.7798 | 4.6532 | - | |
| CoNNear$_{IHC-ANF}$ | 5,066,308 | 819.2 | 11.8147 | 0.0676 | 803.48 | 16.61 |
| Cochlea–IHC–ANF model | - | 819.2 | 167.4808 | - | - | |
| CoNNear$_{cochlea-IHC-ANF}$ | 16,756,292 | 819.2 | 12.6016 | - | 805.83 | - |

Comparison of the average time required to calculate each stage of the reference and the CoNNear model on a CPU (Intel Xeon E5-2620) and a GPU (Nvidia GTX 1080). For each of the separate stages, the reported time corresponds to the average time needed to process a fixed-size input of $N_{CF} = 201$ frequency channels (population response) and $N_{CF} = 1$ channel (single-unit response), corresponding to the output of the preceding stage of the analytical model to a speech stimulus. The same results are shown for the CoNNear IHC–ANF complex, after connecting all the individual modules. The last row shows the computation time needed to transform a speech window input to ANF firing rates, after connecting the CoNNear cochlea and IHC–ANF modules together.

Fig. 2 in ref. [61]) was also reflected in the ANF level-curves, which showed compression at lower stimulus levels for the 1-kHz CF.

Lastly, our CoNNear ANF architectures captured ANF synchrony-level curves well, while showing no apparent differences between the different non-linearities (Fig. 6b). In qualitative agreement with the reference experimental data, the maxima of the synchrony-level curves shifted towards higher levels as the fibers' threshold and rate-level slope increased. At the same time, enhanced synchrony for LSR over HSR fibers was observed for medium to high stimulus levels, with the most pronounced difference for the 1-kHz simulations (dashed lines). For MSR and LSR fibers, the CoNNear models were able to simulate modulation gain, i.e., vector strength $>0.5$[51]. Taken together, we conclude that CoNNear$_{IHC-ANF}$ simulates the properties of IHC and ANF processing on six classical neuroscience experiments well, even though the stimuli for these experiments were unseen, i.e., not presented during the training procedure.

**CoNNear as a real-time model for audio applications.** Due to its CNN architecture, the CoNNear IHC–ANF computations can be sped up when run on an AI accelerator (GPU, VPU etc.). Table 2 summarises the computation time required to execute the final CoNNear architectures on a CPU and GPU, for 201-channel and single-channel inputs. The average computation time is shown for each separate module of the IHC–ANF complex and the respective input length, as well as for the merged IHC–ANF model (CoNNear$_{IHC-ANF}$) after connecting all the separate modules together (see Methods for more information). Lastly, our previously developed CoNNear cochlear model[66] was connected with CoNNear$_{IHC-ANF}$ to directly transform acoustic speech inputs to ANF firing rates.

We did not observe a processing time benefit when running the IHC–ANF stages with 201-channel inputs on a CPU: the CoNNear ANF models actually increased the computation time compared to when the reference models were computed on the same CPU. However, the execution of the 201-channel IHC–ANF models reduced the computation time 12-fold on a GPU, when compared to the execution time of the reference model on the CPU. Our modular design choice makes it possible to use CoNNear$_{IHC-ANF}$ modules only for a subset of CFs, or for single-unit responses. A significant speed up was seen in this latter case: an almost 70-fold faster CPU computation, and a 280-fold speed up for GPU computation. When connected to CoNNear$_{cochlea}$ (Supplementary Fig. 4), ANF firing rates can be simulated in ~800 ms on a CPU and in less than 20 ms on a GPU for an audio input of ~820 ms.

Similar speed-up benefits were observed when simulating single-unit responses for longer inputs on the CPU (Supplementary Table 1), with the ANF models providing a somewhat faster execution. The achieved speed up was more significant on the GPU, with a 1600 times faster CoNNear$_{IHC}$ execution than the reference IHC model and a ~550 times faster CoNNear$_{ANF}$ model execution on average. Different from Table 2, the single-channel CoNNear$_{IHC-ANF}$ models were used for all simulations of longer inputs to avoid large memory allocation. This resulted in lower speed-up benefits for the population-response simulations of Supplementary Table 1, but, for high-end systems that can support this additional memory requirement, the parallel simulation of all $N_{CF}$ channels can provide a speed-up benefit comparable to the results of Table 2.

**Extension of the framework to other auditory model descriptions.** To demonstrate that the presented neural-network framework is applicable to different Hodgkin–Huxley and diffusion-store auditory model desciptions, we used our CNN-based architectures to approximate two other state-of-the-art descriptions of the IHC–ANF complex with varying levels of complexity (see Methods). First, we applied our CoNNear$_{IHC}$ architecture and training approach to approximate the analytical Dierich et al. IHC model description[10]. Figure 7a shows that the trained CNN model (Dierich2020-CNN$_{IHC}$) accurately simulated the steady-state responses of this detailed IHC description (Dierich2020-IHC), as reflected by the AC/DC ratio (Metric 2). However, a property that was not fully captured by our architecture was the adaptation time course of the responses after stimulus onset, as shown in Fig. 7b. The Dierich et al. IHC model requires ~37.5 ms $(t_2 - t_1)$ to decay to its steady-state response while the architecture we used can only accommodate adaptation time course in the order of ~5 ms (Eq. (4)). The higher number of non-linearities and longer time constants that comprise this analytical model (i.e., 7 conductance branches in the Hodgkin–Huxley model) require further adjustments to the CNN architecture (see Discussion for more details). For comparison purposes, the simulated AC/DC ratios of CoNNear$_{IHC}$ and another IHC analytical description, i.e., the Zilany et al. model[58], are also shown in Fig. 7a alongside the experimental data.

Furthermore, we applied the training approach of CoNNear$_{ANF}$ to approximate the instantaneous firing rates of the HSR, MSR,

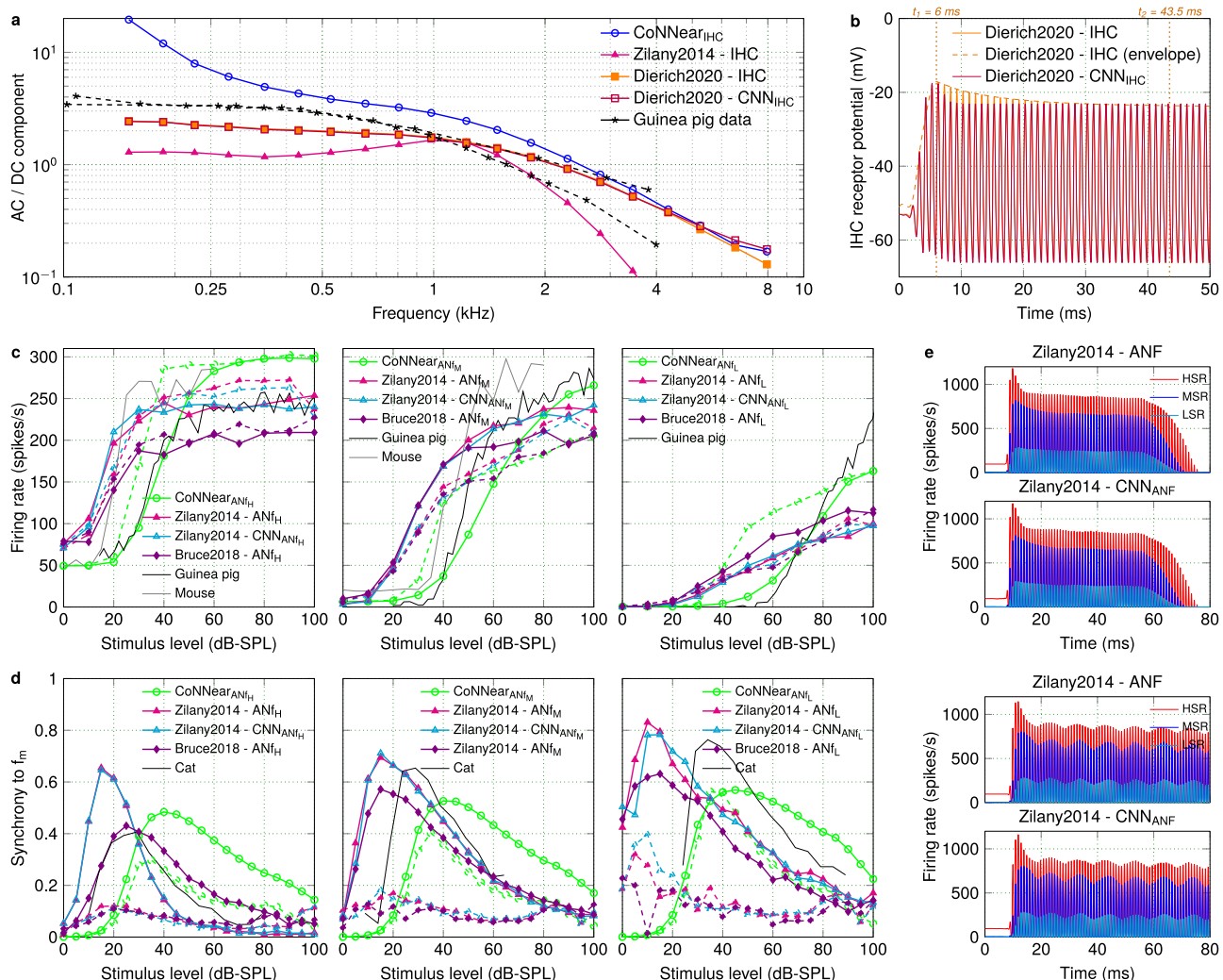

**Fig. 7 Comparison between different IHC–ANF analytical and CNN model descriptions. a** The ratio of the AC and DC components of the IHC responses (Metric 2) is compared across CF between different IHC analytical descriptions[10,58] and the trained CNN models, as well as against guinea-pig data[63]. **b** The IHC receptor potential output of the CNN approximation is compared against the reference Dierich et al. IHC model[10], in response to a 1-kHz pure tone of 70 dB SPL. **c** Rate-level curves (Metric 5) were calculated for the HSR, MSR, and LSR models of different ANF analytical descriptions[58,59] and the CNN models, in response to tone bursts at CFs of 1 (dashed lines) and 4 kHz (solid lines). The experimental data came from guinea-pig[48] and mouse[64] recordings. **d** Synchrony-level functions (Metric 6) were calculated for the HSR, MSR, and LSR models of different ANF analytical descriptions[58,59] and the CNN models, in response to modulated tones with carrier frequencies of 1 (dashed) and 4 kHz (solid) presented at CF. The experimental data came from cat ANF recordings[51]. **e** For each fiber type, the ANF mean firing rate outputs (Metric 4) of the CNN approximations are compared against the reference Zilany et al. ANF model[58], in response to a 1-kHz tone burst and a 1-kHz SAM tone of 70 dB SPL ($f_m = 100$ Hz).

and LSR fiber models included in the Zilany et al. AN analytical description[58], without including the additive Gaussian noise and subsequent spike generator stages of this description. Modelling the schocastic processes of ANF spikes on the basis of post-stimulus time histogram (PSTH) predictions is beyond the scope of the present study. However, after training the CNN models, the generated instantaneous firing rates can be used as input to a spike generator to simulate spike times. The trained CNN models accurately approximated mean firing rates of the analytical ANF models, as shown in response to different tonal stimuli (Fig. 7e and Supplementary Fig. 5b). With the predicted outputs given as inputs to the spike generator model, the simulated PSTH responses were used to compute the ANF rate- and synchrony-level curves (Metrics 5–6) of the different types of ANFs (Fig. 7c, d). The predicted curves (Zilany2014-CNN$_{ANF}$) show a similar trend to the Zilany et al. ANF model (Zilany2014-ANF); however, it is not possible to directly compare the resulting curves due to the inherent noise of the spike generator included in the reference

analytical model. Once again, the simulated rate- and synchrony-level curves are also shown in Fig. 7c, d for CoNNear$_{ANF}$ and for another state-of-the-art ANF description, the Bruce et al.[59] model. In conclusion, our CoNNear architectures extended well to other analytical auditory models with a similar complexity, but need to be adjusted to accommodate the properties of models with higher complexity or longer time constants.

**CoNNear applications**. An important benefit of CNN models over their respective analytical descriptions is given by their dif-ferentiable character. As a result, backpropagation algorithms can be computed from the outputs of these models to train new neural networks. An example use case is presented in Fig. 8a, where a DNN model was trained to minimise the difference between the outputs of two IHC–ANF models: a normal and pathological model. Each model comprised the CoNNear$_{IHC}$ and CoNNear$_{ANF_H}$ modules, and the firing rates of each model were

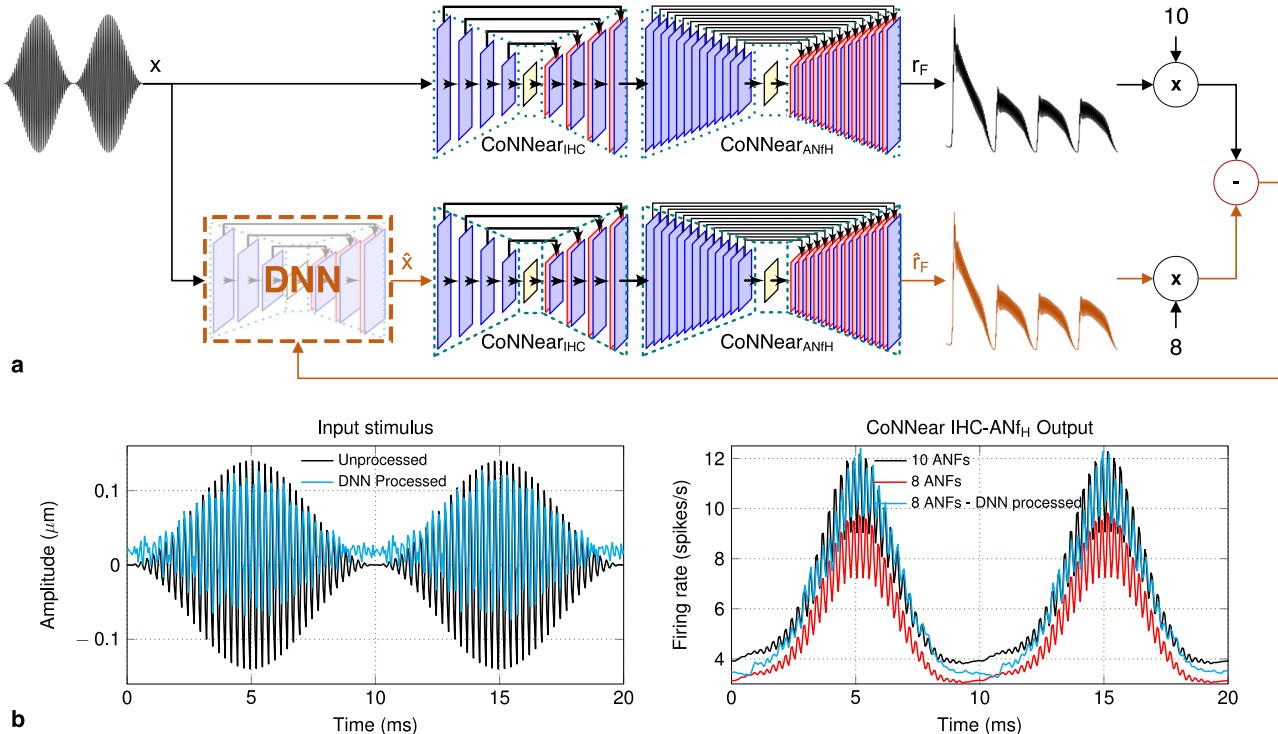

**Fig. 8 Using CoNNear for backpropagation. a** The audio-signal processing DNN model is trained to minimise the difference between the outputs of the two CoNNear IHC–ANF models (orange pathway). **b** When processed by the trained DNN model, the input stimulus results to a firing rate output for the second model that closely matches the firing rate of the first model.

multiplied by a factor of 10 and 8, respectively, to simulate innervations of a normal-hearing human IHC at 4 kHz (Fig. 5 in ref. [79]), and a pathological IHC that has a 20% fiber deafferentation due to cochlear synaptopathy[69]. The DNN model was trained based on the responses of these two CoNNear models to modify the stimulus such to restore the output of the pathological model back to the normal-hearing model output. Training was done using a small input dataset of 4 kHz tones with different levels and modulation depths, normalised to the amplitude ranges of IHC inputs, and the DNN model was trained to minimise the L1-loss between the time and frequency representations of the outputs. After training, the DNN model provides a processed input $\hat{x}$ to the 8-fiber model to generate an output $\hat{r}_F$ that matches the normal-hearing firing rate $r_F$ as much as possible. The result for a modulated tone stimulus is shown in Fig. 8b, for which the amplitude of the 8-fiber model response is restored to that of the normal-hearing IHC–ANF. This example demonstrates the backpropagation capabilites of our CNN models and can, in future studies, be extended to more complex datasets. For instance, the same method could be applied to train on a speech corpus and derive suitable signal-processing strategies for speech processing restoration in hearing-impaired auditory peripheries.

## Discussion

CoNNear presents a new method for projecting complex mathematical descriptions of auditory neuron and synapse models to DNN architectures, while providing a differentiable solution and accelerated run-time. While hybrid approaches have in past studies focussed on optimising analytical model descriptions to reduce their complexity and computation effort[36–38], our method takes advantage of deep learning to develop DNN-based descriptions that can be used for backpropagation in closed-loop systems. Our proposed framework was applied to different auditory Hodgkin–Huxley neuron and synapse models, providing

a baseline methodology that can be extended to more complex biophysical models of sensory systems. The presented CoNNear$_{IHC-ANF}$ model accurately simulates single-unit responses, speeds up the IHC–ANF processing time, and can simulate population responses across a number of simulated tonotopic locations (default $N_{CF} = 201$) when connected to a cochlear model.

**Framework**. The general methodology we followed to model each auditory processing step can be summarised as follows: (i) Derive an analytical description of the biophysical system using available neuroscience recordings. (ii) Use this analytical description to generate a training dataset that contains a broad and representative set of sensory stimuli. (iii) Define a suitable DNN architecture and determine initial values of its hyperparameters based on the properties of the analytical model. (iv) Train the architecture to predict the behaviour of the biophysical system and evaluate its performance on known physiological characteristics using unseen data. A broad set of evaluation metrics derived from experimental neuroscience is used to characterise the physiological aspects of the biophysical system. (v) Adjust and optimise the architecture and its hyperparameters on the basis of the evaluation outcomes. Apart from requiring a careful design of the DNN architecture and a broad range of sensory input stimuli, steps (iii) to (v) need to be repeated to optimise the architecture iteratively and yield a maximally generalisable DNN model.

Here, we focussed our development on creating CNN architectures that optimally approximate the Verhulst et al. IHC–ANF description[61], and showed how these architectures can be optimised based on the specific parameters of each analytical model and the evaluation outcomes. We demonstrated that the proposed framework and CNN architectures generalise well to unseen stimuli, and we used these architectures to show that other auditory sensory cell and synapse model descriptions of

similar complexity can be approximated using the same methodology. This provides a promising outlook because it suggests that our DNN-method might be applicable to other neuronal systems that depend on non-linear and/or coupled ODEs (e.g., see also the application of this method to cochlear mechanics[66]). To further support this claim, we provide a simple example in Supplementary Fig. 6 where our methodology was applied to a non-auditory neuron model, the standard Hodgkin–Huxley model[1]. After determining the baseline CNN architectures of each supplementary model, our outlined iterative procedure can be applied to further improve the accuracy of each approximation, e.g., to fully capture the adaptation properties of the Dierich et al. IHC model or to further reduce the RMSEs of the Zilany et al. ANF model predictions (Supplementary Fig. 5).

The results of our fine-tuning approach for a range of auditory models can provide insight as to how an appropriate initial CNN architecture can be selected depending on the characteristics of the biophysical system that needs to be approximated. For the encoder–decoder CNN architectures we used, the adaptation time constants of the reference analytical descriptions (~8 ms for IHC[9], ~60 ms for AN[12]) guided our choice for the number of convolutional layers and the filter lengths. Using Eq. (4), these two hyperparameters can be selected to yield an architecture with a receptive field (RF) length that roughly corresponds to the adaptation time course of the analytical model to be approximated. The adaptation properties of a model can be estimated from the step response to sustained stimulation (i.e., pure tones for the auditory models and step currents for the standard Hodkin–Huxley model), from which the peak-to-steady-state duration of the response can be computed. For example, the ANF reponses of Verhulst et al. require the whole duration of the selected input window to sufficiently decay to their steady-state response (Fig. 4a and Supplementary Table 2). Equation (4) predicts that at least 12 strided layers were necessary in the encoder (24 in total) to yield a RF larger than the input window and fully capture the adaptation time course of the ANF responses. Using the same formula, we estimate that $N = 6$ layers with a filter length $k_n = 16$ or $N = 3$ layers with filter length $k_n = 128$ are necessary in the encoder to accurately approximate the adaptation properties of the Dierich et al. IHC model (Fig. 7b), which require a RF greater than $(t_2 - t_1) \times fs = 750$ samples. On the other hand, the Zilany et al. AN model had a shorter adaptation time course than the selected window and RF size of CoNNear$_{ANF}$ (Supplementary Table 2), so the same architecture was sufficient to approximate all different fiber types included in this model description. Based on the estimated adaptation properties and Eq. (4), we expect that the number of layers in the CNN architecture can be further reduced without compromising the quality of the predictions for this description.

Supplementary Table 2 summarises the characteristics of all the analytical models we approximated along with their respective CNN encoder–decoder architecture properties. It is important to note that the fine-tuning approach we followed to optimise CoNNear resulted in an architecture for CoNNear$_{IHC}$ that has shorter RF duration than the estimated adaptation of the respective IHC analytical description. This demonstrates the importance of this iterative evaluation procedure to derive maximally generalisable models. While the theoretical RF size (Eq. (4)) can provide an initial estimate of the needed architecture and hyperparameters, this step alone does not take into account the effect of the chosen activation functions between the layers or the RF size that is eventually considered by the convolutional filters after training (see also Methods). These latter aspects can influence the number of units (or effective RF) that the architecture effectively uses after training[80]; thus the evaluation procedure is still necessary to further improve the architectures.

Finally, depending on the time course of the response of an analytical model and the selected number of encoder layers $N$, any multiple of $2^N$ samples can be chosen as input window for the training, as long as it captures the full response time course of the analytical model for a broad range of stimulation paradigms. The number of filters in each layer relate to the non-linear level- and frequency-dependent characteristics of the analytical model and can be selected based on the complexity of the description (e.g., number of ODEs). Since 128 filters per layer were sufficient to describe the properties of the transmission-line cochlear model[66], a highly complex and non-linear system, this number should prove a good starting point for approximating different non-linear systems. By examining whether the properties of the model are faithfully captured by the trained CNN architecture across a broad range of stimulus levels and frequencies, the filter size can be further optimised.

**Limitations**. A limitation of DNN-based descriptions is that, when trained, specific model parameters cannot be adjusted based on physiological insight, as in the case of their analytical counterparts. Instead, the parameters of the mechanistic model need to be adjusted and generate a new training dataset that can be used to derive a new DNN model. In a recent study, we showed how transfer learning can be used to speed up the process of retraining parameters of a known CNN model with the same architecture, where CoNNear$_{cochlea}$ was retrained to approximate the pathological (hearing-impaired) output of the reference analytical cochlear model in less than 10 min[67]. We can use the same approach to retrain or further optimise all CoNNear modules on the basis of improved analytical model descriptions or large neural datasets, when these become available. As DNN approaches learn the properties of a biophysical system solely based on input and output data, DNN-based neuronal models could provide new tools for neuroscientists to explain complex neuronal mechanisms such as heterogenous neural activity, circuit connectivity or optimisation, when properly benchmarked[30,33,34].

Approximately three and eigth days were needed to train each CoNNear ANF model and IHC model, respectively. To shorten these training durations, a different scaling of the datasets, or batch normalisation between the convolutional layers, could prove beneficial[81]. When considering the adaptation and recovery properties of the CoNNear ANF models, a compromise was made to limit the computational complexity of the resulting architectures. As shown in Fig. 4a, b, the selected context and input lengths resulted in ANF models that can simulate up to ~400 ms recovery from prior stimulation and up to ~400 ms adaptation to the steady-state response, respectively. Depending on their application, the CoNNear ANF architectures could be extended to train using longer context or input window lengths, but this choice could sacrifice the speed-up benefits of the models while only improving the accuracy by ~4% for sustained stimulation of >400 ms or for >400 ms interstimulus intervals. On the other hand, when considering their use for real-time applications, where ANF adaptation and recovery properties may be of lesser importance, it is possible to further reduce the context and window sizes and bring execution times below 10 ms. In studies where the saturated, steady-state responses are utilised, faster ANF models that are blind to long interstimulus intervals and unable to describe adaptation over time (Supplementary Fig. 2) might be desirable.

The most significant speed-up benefit compared to the analytical model was observed when connecting the previously developed CoNNear$_{cochlea}$[66] to our CoNNear$_{IHC–ANF}$ (Table 2). The reason for this performance difference relates to the higher complexity (and frequency resolution) of the reference

transmission-line cochlear model[60] and the modular approach we adopted for modelling the IHC–ANF complex. Supplementary Table 1 compares how the execution time and associated complexity of each analytical model implementation impacted the achieved speed-up benefit. Although the CoNNear IHC–ANF architectures use considerably less trainable parameters than the CoNNear cochlear model, the same operations are applied $N_{CF} = 201$ times, i.e., for each frequency channel of the BM output. This leads to a larger memory allocation and associated increased computation time when compared to CoNNear$_{cochlea}$, which uses a single channel throughout the architecture and only adds a 201-sized output layer at the end. Thus, our modular modelling approach negatively affected the computation speed of CoNNear$_{IHC–ANF}$, but resulted in a biophysically correct and rather versatile model, where the number of channels can easily be adjusted depending on the application (e.g., cochlear implants with different numbers of electrodes). Even though significant speed-up benefits might not be achieved when approximating computationally efficient analytical models, our framework can still generate differentiable descriptions that can be backpropagated through.

Depending on the number $N$ of strided convolutional layers used in the encoder of each architecture, the input $L_c$ to each specific CoNNear model needs to be a multiple of $2^N$: a multiple of 8 samples for the IHC model and a multiple of 16384 samples for the ANF models. This particularly limits the performance scalability of CoNNear$_{ANF}$ to different input sizes, since shorter inputs need to be zero-padded. Additionally, when simulating longer inputs with $N_{CF} = 201$ frequency channels, it is better to use the single-channel CoNNear$_{IHC–ANF}$ models and simulate each channel consecutively, rather than simultaneously, to avoid large memory allocation in mid-range systems (Supplementary Table 1).

The training material we used (TIMIT speech corpus) had a sampling frequency of 16 kHz; therefore, the trained IHC–ANF models are roughly limited to operating frequencies up to 8 kHz. This effect is demonstrated in Supplementary Fig. 1, where the RMSE of the IHC excitation pattern of our final IHC architecture is significantly higher for an 8-kHz pure tone than for lower frequencies. To extend the operating frequency of our models, training datasets with a higher sampling frequency or broader frequency content could be used and the sampling frequency of the CoNNear models ($f_s = 20$ kHz) could be increased. It should be noted that, when adapting the CoNNear sampling frequency, the input lengths $L$ and filter lengths of each CNN model need to be adjusted accordingly to correspond to the same window durations. This may compromise the real-time capabilities of the models for applications that require low latencies, but may not be an issue for neuroscience studies.

Lastly, the developed CoNNear models are suitable for implementation in data processing devices such as a cochlear implant to provide biophysically accurate stimuli to the auditory nerve. The ANF responses could also be used to drive neural-network back-ends that simulate brainstem processing or even the generation of auditory evoked potentials, such as the auditory brainstem response[60,61] or the compound action potential[82]. All developed CoNNear modules can be integrated as part of brain networks, neurosimulators, or closed-loop systems for auditory enhancement or neuronal-network based treatments of the pathological system. Our framework for auditory neurons and synapses can inspire new neural-network models or large-scale neural networks that advance our understanding of the underlying mechanisms of such systems, while making use of the transformative ability of backpropagating through these large-scale systems. We think that this type of neural networks can

provide a powerful tool to delve deeper into unknown systems of higher processing levels, such as the brainstem, midbrain, and cortical pathway of the human auditory processing.

## Methods

The procedure depicted in Fig. 1a was used to train the CoNNear IHC and ANF modules using simulated responses of an analytical Hodgkin–Huxley-type IHC model[44] and a three-store diffusion model of the ANF synapse[9], respectively. We adopted the implementations described in ref. [61] and found on https://doi.org/10.5281/zenodo.3717431. The choice of using CNN encoder–decoder architectures for our model was made because of their increased efficiency and parallelism compared to other DNN architectures, such as recurrent neural networks (RNNs), that require sequential processing[83]. CNN architectures only rely on convolutions to transform input to output and are able to apply the same filter functions across multiple windows of the input in parallel[18,84]. The same convolutional operations are applied regardless of the size of the input, making these architectures parallelisable and scalable. Recurrent layers such as LSTMs can still be used in connection to CoNNear (e.g., to capture the dependency on prior stimulation without requiring long context windows, or when approximating other systems), but this would lead to sequential systems that are less computationally efficient and unfit for parallel computing.

Figure 1b depicts the CoNNear IHC encoder–decoder architecture we used: an input of size $L_c × N_{CF}$ cochlear BM waveforms is processed by an *encoder* (comprised of three CNN layers) which encodes the input signal into a condensed representation, after which the *decoder* layers map this representation onto $L × N_{CF}$ IHC receptor potential outputs, for $N_{CF} = 201$ cochlear locations corresponding to the filters' centre frequencies. Context is provided by making the previous $L_l = 256$ and following $L_r = 256$ input samples also available to an input of length $L = 2048$, yielding a total input size of $L_c = L_l + L + L_r = 2560$ samples.

The three CoNNear ANF models follow an encoder–decoder architecture as depicted in Fig. 1c: an IHC receptor potential input of size $L_c × N_{CF}$ is first processed by an *encoder* (comprised of $N = 14$ CNN layers) that encodes the IHC input signal into a condensed representation of size $1 × k_N$ using strided convolutions, after which the *decoder*, using the same number of layers, maps this representation onto $L × N_{CF}$ ANF firing outputs corresponding to $N_{CF} = 201$ cochlear centre frequencies. Context is provided by making the previous $L_l = 7936$ and following $L_r = 256$ input samples also available to an input of length $L = 8192$, yielding a total input size of $L_c = L_l + L + L_r = 16384$ samples.

We illustrate the effect of the context window duration in Supplementary Fig. 2, that shows simulated responses of two trained CoNNear ANF$_L$ models to a 8192-sample long 70-dB-SPL speech segment. Considering an architecture with a short context window (c), the simulated response was unable to reach the onset amplitude of the reference LSR fiber model (b) observed for the high CFs at approximately 100 ms (grey dashed box). At the same time, the response for the short-context architecture decayed to a more saturated output after the onset peak, compared to the reference model. In contrast, when using a longer context window, our final CoNNear ANF$_L$ architecture (d) captured the onset peak observed after the long interstimulus interval while showing an unsaturated fiber response that matched the reference model (b). These observations can better be assessed in panels (e) and (f), where the difference between the outputs of the two trained models and the reference LSR model is visualised.

**Training the CoNNear IHC–ANF complex.** IHC–ANF models were trained using reference analytical BM, or IHC, model simulations[61] to 2310 randomly selected recordings from the TIMIT speech corpus[76], which contains a large number of phonetically balanced sentences with sufficient acoustic diversity. The 2310 TIMIT sentences were upsampled to 100 kHz to solve the analytical model accurately[85]. The root-mean-square (RMS) energy of half the sentences was adjusted to 70 dB and 130 dB sound pressure level (SPL), respectively. These levels were chosen to ensure that the stimuli contained a broad range of instantaneous intensities, necessary for the CoNNear models to capture the characteristic input–output and saturation properties of individual IHC[48] and ANFs[78]. The RMS sound intensity of the whole dataset within 4-ms time bins had an average value of $81 ± 33$ dB SPL.

BM displacements, IHC potentials, and ANF firing rates were simulated across 1000 cochlear sections with CFs between 25 Hz and 20 kHz[61]. The corresponding 1000 $y_{BM}$, $V_m$, and ANF$_{h/m/l}$ output waveforms were downsampled to 20 kHz, and only 201 uniformly distributed CFs between 112 Hz and 12 kHz were selected to train the CoNNear models. Above 12 kHz, human hearing sensitivity becomes very poor[86], motivating the chosen upper limit of considered CFs. The simulated data were then transformed into a one-dimensional dataset of $2310 × 201 = 464,310$ different training sequences. This dimension reduction was necessary because the IHC and ANF models are assumed to have CF-independent parameters, whereas the simulated BM displacements have different impulse responses for different CFs, due to the cochlear mechanics[87]. Hence, parameters for a single IHC or ANF model ($N_{CF} = 1$) were determined during training, based on simulated CF-specific BM inputs and corresponding IHC, or ANF, outputs from the same CF. The parameters of the non-linear operations were shared across the time and frequency dimensions (first two dimensions) of the model, i.e., weights were applied only to the filter dimension (third dimension).

For each of the resulting 464,310 training pairs, the simulated BM and IHC outputs were sliced into windows of 2048 samples with 50% overlap and 256 context samples for the IHC model. For the ANF models, silence was also added before and after each sentence with a duration of 0.5 and 1 s, respectively, to ensure that our trained models can accurately capture the recovery and adaptation properties observed in ANF firing rates. The resulting simulated IHC and ANF outputs were sliced into windows of 8192 samples with 50% overlap, using 7936 context samples before and 256 samples after each window.

A scaling of $10^6$ was applied to the simulated BM displacement outputs before they were given as inputs to the CoNNear IHC model, expressing them in [μm] rather than in [m]. Similarly, the simulated IHC potential outputs were multiplied by a factor of 10, expressed in [dV] instead of [V], and a scaling of $10^{-2}$ was applied to the simulated ANF outputs, expressing them in [x100 spikes/s]. These scalings were necessary to enforce training of CoNNear with sufficiently high digital numbers, while maximally retaining the datasets' statistical mean close to 0 and standard deviation close to 1 to accelerate training[81]. For visual comparison between the original and CoNNear outputs, the values of the CoNNear models were scaled back to their original units in all shown figures and analyses.

CoNNear model parameters were optimised to minimise the mean absolute error (L1-loss) between the predicted CoNNear outputs and the reference model analytical model outputs. A learning rate of 0.0001 was used with an Adam optimiser[88] and the entire framework was developed using the Keras machine learning library[89] with a Tensorflow[90] back-end.

After completing the training phase, the IHC and ANF models were extrapolated to compute the responses across all 201 channels corresponding to the $N_{CF} = 201$ tonotopic CFs located along the BM. The trained architectures were adjusted to apply the same calculated weights (acquired during training) to each of the $N_{CF}$ channels of the input, providing an output with the same size, as shown in Fig. 1c. In the same way, the trained models can easily simulate single-CF IHC responses, or be used for different numbers of channels or frequencies than those we used in the cochlear model.

**Evaluating the CoNNear IHC–ANF complex.** Three IHC and three ANF evaluation metrics were used to determine the final model architecture and its hyperparameters, and to ensure that the trained models accurately captured auditory properties, while not overfitting to the training data and generalising to new inputs. The metrics were based on classical experimental neuroscience measurements and together form a comprehensive set of characteristics that describe IHC–ANF processing. Even though any speech fragment can be seen as a combination of basic stimuli such as impulses and tones of varying levels and frequencies, the acoustic stimuli used for the evaluation can be considered as unseen to the models, as they were not explicitly present in the training material.

The evaluation stimuli were sampled at 20 kHz and had a total duration of 128 ms (2560 samples) and 819.2 ms (16384 samples) for the CoNNear IHC model and the CoNNear ANF models, respectively. The first 256 samples of the IHC stimuli and 7936 samples of the ANF stimuli consisted of silence, to account for the respective context of the models. Each time, the evaluation stimuli were passed through the preceding processing stages of the analytical model to provide the necessary input for each CoNNear model, i.e., through the cochlear model for evaluating the CoNNear IHC model and through the cochlear and IHC models for evaluating the CoNNear ANF models.

*Metric 1: IHC excitation patterns.* Experimentally, excitation patterns are hard to construct from IHC recordings, but such patterns can be simulated from the mean IHC receptor potential at each CF in response to tonal stimuli of different levels to reflect the properties of BM processing. Similar to cochlear excitation patterns, IHC patterns show a characteristic half-octave basal-ward shift of their maxima as stimulus level increases[91]. These excitation patterns also reflect the non-linear compressive growth of BM responses with level observed when stimulating the cochlea with a pure tone that has the same frequency as the CF of the measurement site in the cochlea[92].

We calculated excitation patterns for all 201 simulated IHC receptor potentials in response to pure tones of 0.5, 1, and 2 kHz frequencies and levels between 10 and 90 dB SPL using:

$$\text{tone}(t) = p_0 \cdot \sqrt{2} \cdot 10^{L/20} \cdot \sin(2\pi f_{\text{tone}} t), \quad (1)$$

where $p_0 = 2 \times 10^{-5}$ Pa, $L$ corresponds to the desired RMS level in dB SPL, and $f_{tone}$ to the stimulus frequencies. The pure tones were multiplied with Hanning-shaped 5-ms ramps to ensure gradual onsets and offsets.

*Metric 2: IHC AC/DC ratio.* Palmer and Russel recorded intracellular receptor potentials from an IHC in the basal turn of a guinea-pig cochlea, in response to 80-dB-SPL tone bursts[63]. For low-frequency tones, the IHC receptor potential shows a sinusoidal and asymmetrical response compared to the resting potential (Fig. 9 in ref. [63]). As the stimulus frequency increases, responses become more asymmetrical in the depolarising direction with the AC component gradually becoming a fraction of the DC component. To further quantify this observation, Palmer and Russel reported the ratio between the AC and DC response components as a function of stimulus frequency. The AC/DC ratio shows a smooth logarithmic decrease across frequency (mainly observed for frequencies higher than ~600–700 Hz), which has

been ascribed to the properties of the IHC membrane potential[10,44]. As reported in ref. [63], the AC/DC ratio can be used as a metric to characterise synchronisation in IHCs, with higher ratios indicating more efficient phase-locking of the IHC to the stimulus phase.

Our simulations were conducted for 80-ms, 80-dB-SPL tone bursts of different frequencies presented at the respective CFs, and were compared against experimental AC/DC ratios reported for two guinea-pig IHCs. We used a longer stimulus than adopted during the experimental procedures (50 ms), to ensure that the AC component would reach a steady-state response after the stimulus onset. A 5-ms rise and fall ramp was used for the stimuli, and the AC and DC components of the responses were computed within windows of 50–70 ms after and 5–15 ms before the stimulus onset, respectively. For each frequency, the AC/DC ratio was computed by dividing the RMS value of the AC component, defined as the sinusoidal amplitude of the response, by the DC component, defined as the difference between the AC sinusoidal mean value and the resting potential of the response[9].

*Metric 3: IHC potential-level growth.* Capturing the dynamics of outward IHC $K^+$ currents has an important role in shaping ANF response properties of the whole IHC–ANF complex[9,10]. This feature of mechanical-to-electrical transduction compresses IHC responses dynamically and thereby extends the range of $v_{BM}$ amplitudes that can be encoded by the IHC, as postulated in experimental and theoretical studies[8,40]. As the $v_{BM}$ responses only show compressive growth up to levels of 80 dB SPL[61,66], the simulated half-wave rectified IHC receptor potential is expected to grow roughly linearly with SPL (in dB) for stimulus levels up to 90 dB SPL, thus extending the compressive growth range by 10 dB. To simulate the IHC receptor potential, 4-kHz tonal stimuli with levels from 0 to 100 dB SPL were generated, using the same parameters as before (80-ms duration, 5-ms rise/fall ramp). The responses were half-wave rectified by subtracting their DC component, and the RMS of the rectified responses was computed for each level.

*Metric 4: ANF firing rates.* We evaluate key properties of simulated ANF responses to amplitude-modulated and pure-tone stimuli for which single-unit reference ANF recordings are available. We simulated the firing rate for low-, medium-, and high- SR fibers to 1 and 4 kHz tone bursts and amplitude-modulated tones, presented at 70 dB SPL and calculated at the respective CFs. Based on physiological studies that describe phase-locking properties of the ANF[51,65], stronger phase-locking to the stimulus fine structure is expected for the 1-kHz fiber response than for the 4-kHz, where the response is expected to follow the stimulus envelope after its onset. Similar differences are expected for the amplitude-modulated tone responses as well.

Pure-tone stimuli were generated according to Eq. (1) and the amplitude-modulated tone stimuli using:

$$\text{SAM} - \text{tone}(t) = [1 + m \cdot \cos(2\pi f_{\text{mod}} t + \pi)] \cdot \sin(2\pi f_{\text{tone}} t), \quad (2)$$

where $m = 100\%$ is the modulation depth, $f_{mod} = 100$ Hz the modulation frequency, and $f_{tone}$ the stimulus frequency. Amplitude-modulated tones were multiplied with a 7.8-ms rise/fall ramp to ensure a gradual onset and offset. The stimulus levels $L$ were adjusted using the reference pressure of $p_0 = 2 \times 10^{-5}$ Pa, to adjust their RMS energy to the desired level.

*Metric 5: ANF rate-level curves.* Rate-level curves can be computed to evaluate ANF responses to stimulus level changes, in agreement with experimental procedures[48,64]. Rate-level functions were recorded and studied in different mammals, including cat[49], guinea pig[48], gerbil[93], and mouse[64]. Among species, it was observed that the dynamic range of ANF responses across level is strongly affected by the SR of the fiber[64]. Fibers with HSRs show sharp rate-saturation and a small dynamic range, whereas LSR fibers show sloping saturation, or non-saturating, rate-level functions with significantly larger dynamic ranges.

Using Eq. (1), we generated pure-tone stimuli (50-ms duration, 2.5-ms rise/fall ramp) with levels between 0 and 100 dB and frequencies of approximately 1 and 4 kHz, based on the corresponding CFs of the ANF models (1007 and 3972.7 Hz). The rate-level functions were derived by computing the average response 10–40 ms after the stimulus onset (i.e., excluding the initial and final 10 ms, where some spike intervals may include spontaneous discharge[64]). Data from the experimental studies are plotted alongside our simulations and reflect a variety of experimental ANF rate-level curves from different species and CFs.

*Metric 6: ANF synchrony-level curves.* Responses to modulated tones as a function of stimulus level have also been measured in different mammals (e.g., guinea pig[94], chinchilla[95], cat[51], gerbil[96]). Although rate-level curves to SAM tones show small differences to those obtained to unmodulated carrier stimuli[51,96], differences between fiber types have been exhibited in the synchronisation of the ANF responses to the envelope of SAM stimuli. Lower SR fibers synchronise more strongly to the modulation envelope than HSR fibers and at higher sound intensities close to comfortable listening levels[51,97]. At the same time, LSR fibers show a larger dynamic range where significant synchronisation is present[51,94], supporting the hypothesis that fibers with lower SRs are important for hearing at high levels.

Computationally, synchrony to the stimulus envelope can be quantified using the synchronisation index or vector strength[65,98]. Fully modulated 400-ms long

pure tones with a modulation frequency $f_m$ of 100 Hz[51] and carrier frequencies of 1007 and 3972.7 kHz (henceforth referred to as 1 and 4 kHz) were simulated using Eq. (2), and the synchrony-level functions were calculated by extracting the magnitude of the $f_m$ component from the Fourier spectrum of the fibers' firing rate. The $f_m$ magnitude was normalised to the DC component (0 Hz) of the Fourier spectrum, corresponding to the average firing rate of the fiber[65]. Experimental synchrony-level functions[51] show a non-monotonic relation to the stimulus level and exhibit maxima that occur near the steepest part of ANF rate-level curves.

*Root-mean-square error.* The RMSE was predicted between the evaluation metrics to better quantify the accuracy of each trained architecture and to optimise its hyperparameters. The computed error has the same units of measurement as the estimated quantity (expressed in [mV] for the IHC responses and in [spikes/s] for the ANF firing rates), thus it can be directly compared to the evaluation results of the quantified metric to estimate the accuracy of each architecture. For each metric, the RMSE was computed between the outputs of the reference model and each CoNNear model using:

$$\text{RMSE} = \sqrt{\frac{\sum_{i=1}^{N}(x_i - \hat{x}_i)^2}{N}}, \tag{3}$$

where $i$ corresponds to each sample number, $N$ to the number of samples, $x_i$ to each sample of the reference model simulated results, and $\hat{x}_i$ to each sample of the respective CoNNear model results.

**Connecting the different CoNNear modules.** We considered the evaluation of each CoNNear module separately, without taking into account the CoNNear models of the preceding stages and thus eliminating the contamination of the results by other factors. Each time, the evaluation stimuli were given as inputs to the reference analytical model of the auditory periphery and the necessary outputs were extracted and given as inputs to the respective CoNNear models. However, the different CoNNear models can be merged together to form different subsets of the human auditory periphery, such as CoNNear$_{\text{IHC–ANF}}$ or CoNNear$_{\text{cochlea–IHC–ANF}}$, by connecting the output of the second last layer of each model (before cropping) to the input layer of the next one. This coupling of different modules can show how well these models work together and whether potential internal noise in these neural-network architectures would affect the final response for each module. Using a CNN model of the whole auditory periphery (Supplementary Fig. 4), population responses can be simulated and additional DNN-based back-ends can be added in future studies to expand the pathways and simulate higher levels of auditory processing.

**Evaluating CoNNear execution time.** A TIMIT speech utterance of 3.6 s was used for the evaluation of the CoNNear execution time, and served as input to the analytical model[61] to simulate the outputs of the cochlear and IHC stages. The cochlear BM outputs were then framed into windows of 2560 samples (102.4 ms) to evaluate the CoNNear IHC model and the IHC outputs into windows of 16384 samples (819.2 ms) to evaluate the CoNNear ANF models. The execution time of each stage was estimated by computing the average time required to process all the resulting windows (Table 2). The same sentence was used to evaluate the total execution time required to transform the auditory stimulus to ANF firing rates using each successive stage of the reference and CoNNear models (Supplementary Table 1). Due to their convolutional nature, our CoNNear architectures are parallelisable, making the performance results scalable, since the same operations are applied for any input length.

Evaluation of the CoNNear execution time was performed on a computer with 64 GB of RAM, an Intel Xeon E5-2620 v4 @ 2.10GHz CPU and a Nvidia GTX 1080 Ti 12GB GPU. All stages of the reference auditory model[61] were implemented in Python, except for the cochlear stage where a C implementation was used for the tridiagonal matrix solver. Thus, both IHC–ANF implementations use general-purpose frameworks which were not optimised for CPU/GPU computing, i.e., Python for the reference models and Keras-Tensorflow-Python for the CoNNear models, to allow for a fair comparison. The CoNNear models were developed and evaluated using Keras and Tensorflow v1. We observed an even larger speed-up when executing the models using Tensorflow v2 (non-quantified) and we expect additional improvements when using dedicated CNN platforms.

**Applying the framework to other analytical models.** To attest the extension of our method to other analytical auditory model descriptions, our framework was first applied to approximate the Dierich et al. IHC description[10]. The cochlear responses of the Verhulst et al. model to the speech sentences used for training CoNNear$_{\text{IHC}}$ were used as inputs to the Dierich et al. model, and the same CNN architecture was trained with the new datasets. The AC/DC ratios of Fig. 7a were computed from the simulated IHC responses of the reference model and the trained CNN approximation, using the tonal stimuli of Metric 2. The same stimuli were used to simulate the AC/DC ratios of the Zilany et al. IHC description[58], given as inputs to its cochlea–IHC model. In line with CoNNear, a sampling frequency of 100 kHz was used for all analytical auditory models and 20 kHz for all trained CNN models.

To approximate the three ANF descriptions included in the Zilany et al. analytical model[58], the speech sentences we used for training CoNNear were first used as inputs to its cochlea–IHC module to extract the IHC potential responses. Subsequently, the IHC outputs were given as inputs to its ANF module to extract the instantaneous firing rates for each fiber type. The implementation of the ANF module was adapted so that the additive Gaussian noise and the spike generator stages included in the Zilany et al. model can be omitted and the mean firing rates can be used for the training datasets. The spike generator module was decoupled from the ANF implementation and then used separately to generate the PSTH responses and simulate the rate- and synchrony-level functions of the trained CNN approximation (Fig. 7c, d). The rate- and synchrony-level curves of the reference Zilany et al.[58] and Bruce et al.[59] ANF models were directly computed from their post-stimulus time histogram (PSTH) responses using 100 stimulus repetitions. The auditory stimulus set, described in the corresponding subsections of Metrics 5 and 6, was used as input to each respective cochlea–IHC and ANF description.

The baseline CoNNear architectures were sufficient for approximating the properties of similar state-of-the-art IHC–ANF analytical models, but different architectures might be necessary to approximate other sensory systems or more complex auditory descriptions. Depending on the characteristics of the analytical model that needs to be approximated, an initial estimation of the required CNN architecture can be derived from the adaptation time properties that characterise the response of the analytical model to sustained stimulation. The receptive field (RF) of a selected CNN architecture can be computed[99] to estimate whether the adaptation time course of a reference model can be sufficiently captured by the architecture. Assuming a CNN encoder–decoder architecture with $N$ layers in the encoder, the RF size $r_N$ of the architecture is given by the summation of the RF sizes of all encoder layers:

$$r_N = \sum_{n=1}^{N}\left((k_n - 1) * \prod_{i=1}^{n-1} s_i\right) + 1, \tag{4}$$

where $k_n$ is the filter length of each layer $n$ and $s_i$ is the stride of the previous layers $i$. In our case, the same filter length and stride was used for each layer of the architectures, so the variables $k_n$ and $s_i$ were constant. While the above formula yields the maximum RF an architecture has access to, a given neuron or layer can be more affected by input units near the center of the RF and may not actually train to use all of the RF[80]. As a result, it is common practise for the maximum RF of a network to even exceed the input dimensions to capture long time dependencies[100]. The set of units that effectively influence a specific architecture is called the *effective* RF and can be estimated by backpropagating a gradient signal from the output layer to the input through the network[80]. A better understanding of the relative importance of units in a CNN architecture, paired with an accurate procedure to estimate the effective RF of an architecture[80,99], will yield even better-informed decisions of the initial hyperparameter selection.

Finally, to demonstrate how our framework can be extended to non-auditory mechanistic models, we applied our training procedure to approximate the standard Hodgkin–Huxley (HH) model for the squid giant axon[1]. The implementation of the Python repository of Kramer et al.[101] was adopted for this purpose. To generate the training datasets, a set of step-like pulse stimuli with randomly selected amplitudes from 0 to 1000 μA/cm$^2$ was given as input to the HH model to simulate the voltage outputs. Using the resulting datasets, we initially trained the same architecture that we used for CoNNear$_{\text{IHC}}$ (Table 1), but found that the number of encoder–decoder layers was insufficient to fully transform step-like inputs to spike trains over the whole time window. Currents ⩾7.5 μA/cm$^2$ can generate an infinite train of spikes, so we needed an architecture with a total RF that exceeds the input window to capture the full spike generation across time for high current levels. Thus, we changed the number of layers to 9, the filter lengths to 16 and the number of filters per layer to 64, as shown in Supplementary Table 2.

To establish an empirical relationship between the adaptation time course of each analytical model and the final CNN architecture, we summarised the estimated adaptation time properties, RF and hyperparameters in Supplementary Table 2. For all auditory models, the reported durations reflect the time that the envelope of the model response needed to decay to its steady state when stimulated with a 1-kHz pure tone of 70 dB SPL (passed through each preceding auditory stage). When choosing the stimulus, we ensured that the resulting response could realistically reflect the adaptation properties of the model, and that the estimated adaptation time course did not increase significantly when using different stimulus levels or frequencies. Depending on the characteristics of each model, the adaptation times can be level- or frequency-dependent, thus the longest duration needs to be considered when determining the model architecture.

**Reporting summary**. Further information on research design is available in the Nature Research Reporting Summary linked to this article.

## Data availability
The source code of the auditory periphery model v1.1 used for training is available via https://doi.org/10.5281/zenodo.3717431or https://github.com/HearingTechnology/Verhulstetal2018Model, the TIMIT speech corpus used for training can be found online[76]. The source data underlying all graphs presented in the main and supplementary figures are available from the CoNNear IHC-ANF model repository https://github.com/

HearingTechnology/CoNNear_IHC-ANF. Figures 2, 3, 5, 6 and 7 in this paper can be reproduced using this repository.

## Code availability

The code for running and evaluating the trained CNN models, including instructions of how to execute it, is available via https://doi.org/10.5281/zenodo.4889696 or https://github.com/HearingTechnology/CoNNear_IHC-ANF. A non-commercial, academic UGent license applies.

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

## Acknowledgements

This work was supported by the European Research Council (ERC) under the Horizon 2020 Research and Innovation Programme (grant agreement No. 678120 RobSpear).

## Author contributions

F.D.: conceptualisation, methodology, software, validation, formal analysis, investigation, data curation, writing: original draft, visualisation; D.B.: conceptualisation, methodology; S.V.: conceptualisation, resources, supervision, project administration, funding acquisition, writing: review, editing.

## Competing interests

The authors declare the following competing interests: A patent application (PCTEP2020065893) was filed by UGent on the basis of the research presented in this manuscript. Inventors on the application are S.V., D.B., F.D. and Arthur Van Den Broucke.
