## [Peer Review File · Communications Biology]

Reviewers' Comments:

Reviewer #1:

Remarks to the Author:

Paper presents a technique for creating a DNN approximating the effects of the cochlea and inner hair cells, providing a mechanism for translating auditory input to spike trains which can be computed much more rapidly than by using the more normal detailed computational neuroscience model. This is useful (i) because it's more efficient for simulation on a digital computer (particularly one with a GPU), (ii) because the model is essentially differentiable (which the CN model isn't), and (iii) because it is not tied to a specific model of the cochlea and IHC. This means that it could also in theory be used to develop a hearing aid algorithm for specific patterns of hearing loss (if they were understood in a detailed enough way, and if the hearing aid processor were powerful enough).

There model is available on Github, and the datasets used are public: sufficient information is provided for a researcher to repeat the work.

Overall, this is a very clearly written and well thought out paper which I can't think of any improvements to. Whether this is because it's simply an excellent paper, or whether I'm getting soft in my dotage I can't tell.

I think it is publishable as it is.

Reviewer #2:

Remarks to the Author:

Summary:

This paper presents a model called CoNNear, which employs a CNN with a convolutional encoder-decoder architecture, for the auditory analytical model. This model consists of 3 stages: 1) cochlear processing, 2) IHC transduction and 3) ANF firing. In this paper, the IHC transduction and ANF firing are discussed (the cochlear-processing module is described in other papers). The data to train the models were obtained by analytical models. This resulted in one model for IHC transduction and three models for different ANF types (low, medium and high spiking). It is discussed how the network is built and how the hyperparameters are chosen. Then, the accuracy is discussed based upon plots of the output of the modules. Besides the results in the accuracy, the computational performance was discussed, which showed 70x speedup against the reference model when using a CPU and 280x speedup when using a GPU. After this, the generalizability of the framework is shown and a use case is presented. Finally, the results are discussed and concluded.

Positive aspects:

1. The paper presents a use case which really makes clear what the practical benefit of the presented models are comparing to the analytical ones. This use case shows that, because of the backpropagation, a pathological sound signal can be resolved. This can be useful in hearing-impaired cochlea.
2. The description of the architecture is elaborate. Furthermore, it is shown how different parameters of the model influence the output, making it understandable why they are chosen for the parameters in the case of auditory model.
3. The availability of the data and code (which is documented) makes it easier to reproduce the results and use the framework for future work.

Suggestions for improvement:

1. It is hard to judge how accurate the model is. There are shown plots, which show similar behavior to the analytically obtained data. There were mentioned about six evaluation metrics, some of which are used in the plots. However, these results were not compared to other state-of-the-art models, consequently, for someone who does not know the current state of the art in auditory sensory-cell models, it is hard to judge how accurate the presented models are. Additionally, the metrics were vaguely discussed; for example: "generally followed the pure-tone

excitation patterns of the reference model (a), but showed a rather noisy response 164 across CFs, especially for the higher stimulation levels." And "This combination of non-linearities (d) compressed the responses of the CoNNear(ANfH) and CoNNear(ANfM) models even more, and negatively affected the onset responses." These kinds of descriptions do not give the reader a good impression what characteristics the model can simulate and what its limitations are. Giving a clear overview of how well this approach scores on the evaluation metrics in comparison to other state-of-the-art models could give the reader a good idea.

2. It is hard to judge from the paper how well the model generalizes for multiple neural models. In the text is mentioned that "Our general framework for modelling sensory-cells ad synapses ..." and "All the developed CoNNear architectures can easily be integrated as part of brain networks, neurosimulators, or closed-loop systems for auditory enhancement or neuronal-network based treatments of the pathological system. Further neural network models can be developed on the basis of the present framework to compose large-scale neuronal networks and advance our understanding of the underlying mechanisms of such systems ...". This implies that this approach works for all Hodgkin-Huxley like models. However, in the paper it is not discussed how high the accuracy is expected to be for different kinds of models and the reader is not guided to how to choose the parameters of the DNN architecture. This is strengthened by the fact that, in the section where generalizability is discussed, only auditory neurons and synapses are discussed. Moreover, it is also mentioned that "our method might be applicable to other neuronal systems that depend on nonlinear and/or coupled ODEs", which is weaker than the two previously mentioned statements. The authors are advised to make the text clearer about what is meant with generalizability of the framework.

3. A critical aspect of the work in this paper is the use of a CNN with a convolutional encoder-decoder architecture. However, it is not mentioned why this architecture is chosen and not for example RNNs. This while it is mentioned in the text that "A different approach could be the use of recurrent layers (e.g., LSTM) within the CoNNear architectures to capture the dependency to prior stimulation without requiring long context windows". It would be interesting if a small discussion on the architecture would be added to the paper so that the reader can better judge whether this framework can be beneficial for future work and why CNNs over other models have been preferred.

4. The higher performance of the presented method in this paper is one of its selling points. However, it is not mentioned, how well-optimized the reference optimal implementation is and in which language this implementation is made. (It is mentioned in the methods that the Keras machine learning library with TensorFlow is used for the CNNs). Both factors can have a significant influence on the performance and can make the speedup number more or less impressive. Furthermore, the scaling of the performance numbers is not discussed. The scaling will be interesting, especially given that generalizability is advertised, as that gives the reader an idea how well this approach will perform for other neural models. Consequently, the authors are advised to improve the performance section so it will be clearer how good the presented work is.

5. The Discussion is just a summary of the paper and does not clearly describe the limitations. Consequently, it is hard to quickly judge how good the work is and if it should be used as there is not a clear overview compared against other related work.

6. It is mentioned that the quality of the CNN models is evaluated based on difficult properties (i.e., AC/DC ratio, excitation patterns, ANF firing rate, rate-level curves and modulation synchrony). However, these properties are not explained to the reader. The reader is advised to introduce these properties to the reader as it will make the presented work easier understandable. The same statements hold for the presented variables such as $N(\text{CF})$, which was used without explanation.

All in all, this is a solid piece of work. However, it requires some added information to be complete. As such, a revision to provide added information is recommended.

Response to review of manuscript COMMSBIO-20-3661A

A neural-network framework for modelling auditory sensory cells and synapses
Fotios Drakopoulos, Deepak Baby, Sarah Verhulst

Dear Reviewers,

We thank you for evaluating our manuscript and for the constructive reviews. We carefully considered your concerns and suggestions for changes, and as you can see in this rebuttal letter, we were able to address almost all of your concerns. We hope you find our revised manuscript improved, and can endorse its publication after reading this letter and the manuscript. Our point-by-point answers to the original rebuttal letter (in black) are given in a blue font. Where indicated, line and page numbers correspond to the clean version of the manuscript.

Yours Sincerely,
Fotios Drakopoulos, Deepak Baby, Sarah Verhulst

Reviewer 1:

Thank you for your very positive feedback, we are humbled to read these kind words.

Reviewer 2:

Thank you for your thorough assessment and feedback. Our point-by-point answers are given below.

1. It is hard to judge how accurate the model is. There are shown plots, which show similar behavior to the analytically obtained data. There were mentioned about six evaluation metrics, some of which are used in the plots. However, these results were not compared to other state-of-the-art models, consequently, for someone who does not know the current state of the art in auditory sensory-cell models, it is hard to judge how accurate the presented models are. Additionally, the metrics were vaguely discussed; for example: “generally followed the pure-tone excitation patterns of the reference model (a), but showed a rather noisy response 164 across CFs, especially for the higher stimulation levels.” And “This combination of non-linearities (d) compressed the responses of the CoNNear(ANfH) and CoNNear(ANfM) models even more, and negatively affected the onset responses.” These kinds of descriptions do not give the reader a good impression what characteristics the model can simulate and what its limitations are. Giving a clear overview of how well this approach scores on the evaluation metrics in comparison to other state-of-the-art models could give the reader a good idea.

To better quantify the accuracy of our IHC-ANF models, root-mean-square errors (RMSEs) were added for the IHC excitation patterns and ANF firing rates (see Supplementary Figs 3 & 4 and RMSE section in Methods). The RMSEs were computed between the reference analytical model (Verhulst2018) and each trained CoNNear architecture and were used together with the evaluation outcomes to determine the final CoNNear architectures that optimally approximate the reference model. The “Determining the CoNNear hyperparameters” section (pages 4-7) was made more clear to justify our architecture optimisation (based on the RMSE results as well), and

the evaluation metrics are now explained in more detail in Methods (lines 502-586). Difference plots were also provided for Supplementary Fig. 1, so that the accuracy of the LSR ANF CoNNear model can also be visually assessed (panel f).

A detailed comparison between our reference analytical model and other state-of-the-art analytical models is out of the scope of this paper, but we can refer to other papers that have compared characteristics of auditory models (Saremi et al. 2016; <https://doi.org/10.1121/1.4960486> or Verhulst et al. 2018; <https://doi.org/10.1016/j.heares.2017.12.018>). The main focus of the paper was to apply our methodology to derive an accurate CNN-based approximation of the Verhulst et al. IHC-ANF model (CoNNear), as we also point out in Discussion: “Here, we focussed our development on creating CNN architectures that optimally approximate the Verhulst et al. IHC-ANF model [61], and showed how these baseline architectures can be adjusted based on the specific parameters of each analytical model and the evaluation outcomes.” However, we did compare how well our CoNNear implementation juxtaposed against two other state-of-the-art IHC-ANF models when using the same evaluation metrics (Fig. 7). We performed this comparison for the analytical models and their respective CNN approximations, and added physiological reference data when available. To provide a quantitative metric for the approximation quality, we also computed RMSEs for the CNN approximations of these two IHC-ANF models (Supplementary Fig. 5).

2. It is hard to judge from the paper how well the model generalizes for multiple neural models. In the text is mentioned that “Our general framework for modelling sensory-cells and synapses ...” and “All the developed CoNNear architectures can easily be integrated as part of brain networks, neuro-simulators, or closed-loop systems for auditory enhancement or neuronal-network based treatments of the pathological system. Further neural network models can be developed on the basis of the present framework to compose large-scale neuronal networks and advance our understanding of the underlying mechanisms of such systems ...”. This implies that this approach works for all Hodgkin-Huxley like models. However, in the paper it is not discussed how high the accuracy is expected to be for different kinds of models and the reader is not guided to how to choose the parameters of the DNN architecture. This is strengthened by the fact that, in the section where generalizability is discussed, only auditory neurons and synapses are discussed. Moreover, it is also mentioned that “our method might be applicable to other neuronal systems that depend on nonlinear and/or coupled ODEs”, which is weaker than the two previously mentioned statements. The authors are advised to make the text clearer about what is meant with generalizability of the framework.

The Discussion section was updated and subdivided into different subsections to clarify the framework’s capabilities. Our framework was applied to the Verhulst et al., Dierich et al. and Zilany et al. IHC-ANF models, as well as to the cochlear model of Verhulst et al. (<https://doi.org/10.1038/s42256-020-00286-8>), to show how this methodology can be used to approximate different auditory models. This attests to the generalizability of our method in the way that the reader can follow the proposed generic formula (“Framework” subsection of Discussion; lines 339-349) to model different kinds of biophysical systems. To strengthen our statements, we provided an additional example by applying our method to the standard Hodgkin-Huxley model (Supplementary Figure 6): “This provides a promising outlook because it suggests that our DNN-method might be applicable to other neuronal systems that depend on nonlinear and/or coupled ODEs (e.g., see also the application of this method to cochlear mechanics [63]). To further support this claim, we provide a simple example in Supplementary Fig. 6 where our methodology was applied to a non-auditory neuron model, the standard Hodgkin-Huxley model [1].”

3. A critical aspect of the work in this paper is the use of a CNN with a convolutional encoder-decoder architecture. However, it is not mentioned why this architecture is chosen and not for example RNNs. This while it is mentioned in the text that “A different approach could be the use of recurrent layers (e.g., LSTM) within the CoNNear architectures to capture the dependency to prior stimulation without requiring long context windows”. It would be interesting if a small discussion on the architecture would be added to the paper so that the reader can better judge whether this framework can be beneficial for future work and why CNNs over other models have been preferred.

Thank you for the suggestion, we added an explicit description at the start of the Methods section (lines 411-419) which motivates our choice of adopting CNNs for the CoNNear architecture: “Our choice of using CNN encoder-decoder architectures for our model was made because of their increased efficiency and parallelism compared to other DNN architectures, such as recurrent neural networks (RNNs), that require sequential processing [81]. CNN architectures only rely on convolutions to transform input to output and are able to apply the same filter functions across multiple windows of the input in parallel [18, 82]. The same convolutional operations are applied regardless of the size of the input, making these architectures much more hardware-friendly and scalable. Recurrent layers can still be used in connection to CoNNear (e.g., to capture the dependency on prior stimulation without requiring long context windows, or when approximating other systems), but this would lead to sequential systems that are less computationally-efficient and unfit for parallel computing”

4. The higher performance of the presented method in this paper is one of its selling points. However, it is not mentioned how well-optimized the reference optimal implementation is and in which language this implementation is made. (It is mentioned in the methods that the Keras machine learning library with TensorFlow is used for the CNNs). Both factors can have a significant influence on the performance and can make the speedup number more or less impressive. Furthermore, the scaling of the performance numbers is not discussed. The scaling will be interesting, especially given that generalizability is advertised, as that gives the reader an idea how well this approach will perform for other neural models. Consequently, the authors are advised to improve the performance section so it will be clearer how good the presented work is.

We have expanded the description in Methods that now has a dedicated “Evaluating CoNNear execution time” section (lines 609-622), where the performance evaluation of the CNN models is discussed. We accordingly adapted the “CoNNear as a real-time model for audio applications” section as well (lines 254-269). It is correct that the achieved speed-up benefit depends on the efficiency of the reference biophysical model and the DNN architecture that is adopted to approximate it, which makes it hard to generalize the performance benefit of the method for any model, especially because of the large variety of NN methods and analytical model implementations (see also the previous comment). The platforms where the models are executed on can also have an influence on the speed-up that can be achieved, but we used general-purpose, non-optimized frameworks to offer a “worst-case” benefit. The corresponding Methods subsection further describes the languages and evaluation methods we used: “Due to their convolutional nature, our CoNNear architectures are parallelisable, making the performance results scalable, since the same operations are applied for any input length. All stages of the reference auditory model [61] were implemented in Python, except for the cochlear stage where a C implementation was used for the tridiagonal matrix solver. Thus, both IHC-ANF implementations use general-purpose frameworks which were not optimized for CPU/GPU computing, i.e., Python for the reference models and Keras-Tensorflow-Python for the CoNNear models, making the comparison between them fair. We expect additional speed-up improvements of CoNNear when using dedicated CNN platforms in the future.”

5. The Discussion is just a summary of the paper and does not clearly describe the limitations. Consequently, it is hard to quickly judge how good the work is and if it should be used as there is not a clear overview compared against other related work.

We agree that the limitations of our method should be more specifically mentioned, and hence we expanded the Discussion section with a “Limitations” subsection (lines 362-394). In summary, we describe limitations related to the optimisation, training and operating limits of the resulting CoNNear models. A direct comparison against other NN-approaches to simulate neurons and synapses is difficult (and perhaps not meaningful) because there is a large variety of possible NN-implementations and none of the existing approaches focus on the approximation of analytical descriptions of auditory neurons and models. As we also mention in Discussion (lines 330-333): “While hybrid approaches have in past studies focussed on optimising analytical model descriptions to reduce their complexity and computation effort [36–38], our method takes advantage of deep-learning to develop DNN-based descriptions that can be used for backpropagation in closed-loop systems.”

6. It is mentioned that the quality of the CNN models is evaluated based on difficult properties (i.e., AC/DC ratio, excitation patterns, ANF firing rate, rate-level curves and modulation synchrony). However, these properties are not explained to the reader. The reader is advised to introduce these properties to the reader as it will make the presented work easier understandable. The same statements hold for the presented variables such as $N(CF)$, which was used without explanation.

We agree that it was not easy to follow the evaluation metrics for readers without an explicit background in auditory neuroscience. For this reason, we described the evaluation metrics better in each subsection of Methods (lines 502-586), with reference to the experimental neuroscience papers. We also attempted to present all variables in the main text, in cases where the definition was missing (e.g., lines 87-88, 98-100, 106-107). We hope that our additions can guide both experienced and inexperienced readers to understand how we objectively evaluated the CoNNear models.

Reviewers' Comments:

Reviewer #2:

None

Revisited review:

A neural-network framework for modelling auditory sensory cells and synapses

Summary:

This paper presents a model called CoNNear, with the use of a CNN with a convolutional encoder-decoder architecture, for the auditory analytical model. This model consists out of 3 stages: 1) cochlear processing, 2) IHC transduction and 3) ANF firing. In this paper the IHC transduction and ANF firing are discussed (the cochlear processing module is described in other papers). The data to train the models were obtained by analytical models, resulting in one model for IHC transduction and three models for different ANF types (low, medium and high spiking). It is discussed how the network is build and how the hyperparameters are chosen. Then the accuracy is discussed based upon plots of the output of the modules. Besides the results the accuracy the computational performance was discussed which showed 70x speedup against the reference model when using a CPU and 280x speedup when using a GPU. After this, the generalizability of the framework is shown and a use case is presented. Finally, the results are discussed and concluded.

Rebuttal Comments:

1. It is hard to judge how accurate the model is. There are shown plots, which show similar behavior to the analytically obtained data. There were mentioned about six evaluation metrics, some of which are used in the plots. However, these results were not compared to other state-of-the-art models, consequently, for someone who does not know the current state of the art in auditory sensory-cell models, it is hard to judge how accurate the presented models are. Additionally, the metrics were vaguely discussed; for example: “generally followed the pure-tone excitation patterns of the reference model (a), but showed a rather noisy response 164 across CFs, especially for the higher stimulation levels.” And “This combination of non-linearities (d) compressed the responses of the CoNNear(ANfH) and CoNNear(ANfM) models even more, and negatively affected the onset responses.” These kinds of descriptions do not give the reader a good impression what characteristics the model can simulate and what its limitations are. Giving a clear overview of how well this approach scores on the evaluation metrics in comparison to other state-of-the-art models could give the reader a good idea.

To better quantify the accuracy of our IHC-ANF models, root-mean-square errors (RMSEs) were added for the IHC excitation patterns and ANF firing rates (see Supplementary Figs 3 & 4 and RMSE section in Methods). The RMSEs were computed between the reference analytical model (Verhulst2018) and each trained CoNNear architecture and were used together with the evaluation outcomes to determine the final CoNNear architectures that optimally approximate the reference model. The “Determining the CoNNear hyperparameters” section (pages 4-7) was made more clear to justify our architecture optimisation (based on the RMSE results as well), and the evaluation metrics are now explained in more detail in Methods (lines 502-586). Difference plots were also provided for Supplementary Fig. 1, so that the accuracy of the LSR ANF CoNNear model can also be visually assessed (panel f).

The additional supplementary figures make it indeed clearer why architecture optimizations are done.

A detailed comparison between our reference analytical model and other state-of-the-art analytical models is out of the scope of this paper, but we can refer to other papers that have compared characteristics of auditory models (Saremi et al. 2016; <https://doi.org/10.1121/1.4960486> or Verhulst et al 2018; <https://doi.org/10.1016/j.heares.2017.12.018>). The main focus of the paper was to apply our methodology to derive an accurate CNN-based approximation of the Verhulst et al. IHC-ANF model (CoNNear), as we also point out in Discussion: “Here, we focussed our development on creating CNN architectures that optimally approximate the Verhulst et al. IHC-ANF model [61], and showed how these baseline architectures can be adjusted based on the specific parameters of each analytical model and the evaluation outcomes.” However, we did compare how well our CoNNear implementation juxtaposed against two other state-of-the-art IHC-ANF models when using the same evaluation metrics (Fig. 7). We performed this comparison for the analytical models and their respective CNN approximations, and added physiological reference data when available. To provide a quantitative metric for the approximation quality, we also computed RMSEs for the CNN approximations of these two IHC-ANF models (Supplementary Fig. 5).

There are indeed made comparisons between the CNN IHC & ANF models and related work. Furthermore, in Methods (which is only placed after the whole discussion of the models) characteristics of the IHC & ANF models are discussed. However, it is not mentioned explicitly what the evaluation metrics are. For example, it is confusing that there are six evaluation metrics are initially mentioned for the IHC-ANF complex but later, in the Methods section, there are 3 metrics defined for the IHC and 3 for the ANF model. Simply stating which are the 6 metrics, before the results are shown, would make it easier for the reader to get a complete overview of the work done and accuracy achieved.

Additionally, in lines 126-128 is mentioned that: *“Table 1 shows the final layouts of all the CoNNear modules we obtained after taking into account: (i) the L1-loss on the training speech material (i.e., the absolute difference between simulated CNN and analytical responses), (ii) the desired auditory processing characteristics, and (iii) the computational load.”* Although indeed different characteristics are shown and discussed, the RMSE is discussed and it can be expected that bigger CNNs require more computations; yet a clear overview is not given to the reader. Instead, it seems they are discussed separately, which again makes it hard for the reader to get a complete overview.

2. It is hard to judge from the paper how well the model generalizes for multiple neural models. In the text is mentioned that “Our general framework for modelling sensory-cells and synapses ...” and “All the developed CoNNear architectures can easily be integrated as part of brain networks, neuro-simulators, or closed-loop systems for auditory enhancement or neuronal-network based treatments of the pathological system. Further neural network models can be developed on the basis of the present framework to compose large-scale neuronal networks and advance our understanding of the underlying mechanisms of such systems ...”. This implies that this approach works for all Hodgkin-Huxley like models. However, in the paper it is not discussed how high the accuracy is expected to be for different kinds of models and the reader is not guided to how to choose the parameters of the DNN architecture. This is strengthened by the fact that, in the section where generalizability is discussed, only auditory neurons and synapses are discussed. Moreover, it is also mentioned that “our method might be applicable to other neuronal systems that depend on nonlinear and/or coupled ODEs”, which is weaker than the two previously mentioned statements. The authors are advised to make the text clearer about what is meant with generalizability of the framework.

The Discussion section was updated and subdivided into different subsections to clarify the framework’s capabilities. Our framework was applied to the Verhulst et al., Dierich et al. and Zilany et al. IHC-ANF models, as well as to the cochlear model of Verhulst et al. (<https://doi.org/10.1038/s42256-020-00286-8>), to show how this methodology can be used to approximate different auditory models. This attests to the generalizability of our method in the way that the reader can follow the proposed generic formula (“Framework” subsection of Discussion; lines 339-349) to model different kinds of biophysical systems. To strengthen our statements, we provided an additional example by applying our method to the standard Hodgkin-Huxley model (Supplementary Figure 6): *“This provides a promising outlook because it suggests that our DNN-method might be applicable to other neuronal systems that depend on nonlinear and/or coupled ODEs (e.g., see also the application of this method to cochlear mechanics [63]). To further support this claim, we provide a simple example in Supplementary Fig. 6 where our methodology was applied to a non-auditory neuron model, the standard Hodgkin-Huxley model [1].”*

The added supplementary figure, showing that besides auditory systems also the standard Hodgkin-Huxley (HH) model can be simulated (at least for a certain input), indeed adds support to the claim of the method to be generalizable. However, it is still missing clear guidance on how the network should be adjusted/ chosen. This as it was only shown that the methodology works also for the HH model, not what changes were made to the CNN. This in combination with when discussing the performance of the method on the model of Dierich et al.: *“The higher number of non-linearities comprised in this analytical model (i.e., 7 conductance branches in the Hodgkin-Huxley model) might require adaptations to the CNN architecture to accommodate this, e.g., by including an additional layer or longer filter durations to yield more accurate simulations (see also Discussion and Supplementary Fig. 5a).”* (line 285-288), and *“Thus, the same iterative procedure can be applied to further improve the accuracy of each supplementary CNN model, e.g., to fully capture the adaptation properties of the Dierich et al. IHC model (see Generalisability) or minimize the RMSEs of the Zilany et al. ANF models (Supplementary Fig. 5).”* (line 359-360). This only gives statements (make the CNN bigger) which are common

practice when applying CNNs, without giving extra insights of how much bigger the CNN needs to be and how this influences both accuracy and computation time.

Furthermore, the framework basically consists of saying that training data should come from analytical models and then use the standard approach of developing a CNN (choosing the parameters of the model). The extra insights on what a good starting point is for the CNN and how complexity of a model influences the accuracy and computational work is missing. This for example shows in line 134-136: *“Prior knowledge of fine-tuning a neural-network-based model of human cochlear processing [63] helped us to make initial assumptions about the needed architecture to accurately capture the computations performed by the analytical IHC model [61].”* This does not give much insight how an architecture should be chosen for new models.

3. A critical aspect of the work in this paper is the use of a CNN with a convolutional encoder-decoder architecture. However, it is not mentioned why this architecture is chosen and not for example RNNs. This while it is mentioned in the text that “A different approach could be the use of recurrent layers (e.g., LSTM) within the CoNNear architectures to capture the dependency to prior stimulation without requiring long context windows”. It would be interesting if a small discussion on the architecture would be added to the paper so that the reader can better judge whether this framework can be beneficial for future work and why CNNs over other models have been preferred.

Thank you for the suggestion, we added an explicit description at the start of the Methods section (lines 411-419) which motivates our choice of adopting CNNs for the CoNNear architecture: *“Our choice of using CNN encoder-decoder architectures for our model was made because of their increased efficiency and parallelism compared to other DNN architectures, such as recurrent neural networks (RNNs), that require sequential processing [81]. CNN architectures only rely on convolutions to transform input to output and are able to apply the same filter functions across multiple windows of the input in parallel [18, 82]. The same convolutional operations are applied regardless of the size of the input, making these architectures much more hardware-friendly and scalable. Recurrent layers can still be used in connection to CoNNear (e.g., to capture the dependency on prior stimulation without requiring long context windows, or when approximating other systems), but this would lead to sequential systems that are less computationally-efficient and unfit for parallel computing”*

The added information about why there is chosen for a CNN is fine. Although, a comparison of RNNs vs. CNNs both in performance and accuracy would certainly be of additional value, but maybe out of scope of this paper.

(Besides, this added value was shown especially in the sentence *“A different approach could be the use of recurrent layers (e.g., LSTM) within the CoNNear architectures to capture the dependency to prior stimulation without requiring long context windows”* which was removed from the latest paper revision).

4. The higher performance of the presented method in this paper is one of its selling points. However, it is not mentioned how well-optimized the reference optimal implementation is and in which language this implementation is made. (It is mentioned in the methods that the Keras machine learning library with TensorFlow is used for the CNNs). Both factors can have a significant influence on the performance and can make the speedup number more or less impressive. Furthermore, the scaling of the performance numbers is not discussed. The scaling will be interesting, especially given that generalizability is advertised, as that gives the reader an idea how well this approach will perform for other neural models. Consequently, the authors are advised to improve the performance section so it will be clearer how good the presented work is.

We have expanded the description in Methods that now has a dedicated “Evaluating CoNNear execution time” section (lines 609-622), where the performance evaluation of the CNN models is discussed. We accordingly adapted the “CoNNear as a real-time model for audio applications” section as well (lines 254-269). It is correct that the achieved speed-up benefit depends on the efficiency of the reference biophysical model and the DNN architecture that is adopted to approximate it, which makes it hard to generalize the performance benefit of the method for any model, especially because of the large variety of NN methods and analytical model implementations (see also the previous comment). The platforms where the models are executed on can also have an influence on the speed-up that can be achieved, but we used general-purpose, non-optimized frameworks to offer a “worst-case” benefit. The corresponding Methods subsection further describes the

languages and evaluation methods we used: “Due to their convolutional nature, our CoNNear architectures are parallelisable, making the performance results scalable, since the same operations are applied for any input length. All stages of the reference auditory model [61] were implemented in Python, except for the cochlear stage where a C implementation was used for the tridiagonal matrix solver. Thus, both IHC-ANF implementations use general-purpose frameworks which were not optimized for CPU/GPU computing, i.e., Python for the reference models and Keras-Tensorflow-Python for the CoNNear models, making the comparison between them fair. We expect additional speed-up improvements of CoNNear when using dedicated CNN platforms in the future.”

It is indeed true that it is hard to give a performance benefit for each model. That is why it would have been nice if there was explained/showed how the CNN model scales for different numbers of input/ different sizes, so the reader can have an indication what the performance will be for different models. Furthermore, it is still not explained on which CPU and GPU the measurements are done. As for the IHC and ANF models, the speedup seems to come from using the GPU, which is indeed possible due to the used method. However, as there are also HH models with analytical GPU implementations it might be that these models would not benefit from using this methodology. Currently, the reader does not gain any information due to this.

Another point of interest is that the best performance gain is showed by the CoNNear_{cochlea-IHC-ANF} model, while the CoNNear_{IHC-ANF} speedup is not that high. Consequently, it looks like the most impressive speedup is achieved with previous work, which is also not explained in this paper so the reader has to guess this.

5. The Discussion is just a summary of the paper and does not clearly describe the limitations. Consequently, it is hard to quickly judge how good the work is and if it should be used as there is not a clear overview compared against other related work.

We agree that the limitations of our method should be more specifically mentioned, and hence we expanded the Discussion section with a “Limitations” subsection (lines 362-394). In summary, we describe limitations related to the optimisation, training and operating limits of the resulting CoNNear models. A direct comparison against other NN-approaches to simulate neurons and synapses is difficult (and perhaps not meaningful) because there is a large variety of possible NN-implementations and none of the existing approaches focus on the approximation of analytical descriptions of auditory neurons and models. As we also mention in Discussion (lines 330-333): “While hybrid approaches have in past studies focussed on optimising analytical model descriptions to reduce their complexity and computation effort [36–38], our method takes advantage of deep-learning to develop DNN-based descriptions that can be used for backpropagation in closed-loop systems.”

The limitations of the method used in this paper are better discussed now. However, it is still a huge limitation that by just reading the paper (and having previous knowledge of the used models) it is really hard to predict how the computation time scales for different problems and what characteristics/properties of neuron models can be modeled efficiently/accurately with the use of a CNN.

6. It is mentioned that the quality of the CNN models is evaluated based on difficult properties (i.e., AC/DC ratio, excitation patterns, ANF firing rate, rate-level curves and modulation synchrony). However, these properties are not explained to the reader. The reader is advised to introduce these properties to the reader as it will make the presented work easier understandable. The same statements hold for the presented variables such as N(CF), which was used without explanation.

We agree that it was not easy to follow the evaluation metrics for readers without an explicit background in auditory neuroscience. For this reason, we described the evaluation metrics better in each subsection of Methods (lines 502-586), with reference to the experimental neuroscience papers. We also attempted to present all variables in the main text, in cases where the definition was missing (e.g., lines 87-88, 98-100, 106-107). We hope that our additions can guide both experienced and inexperienced readers to understand how we objectively evaluated the CoNNear models.

The paper is easier to read regarding the understanding of variables. However, there is still room for improvement as the acronym AN (line 54) is used without introduction. Furthermore, in Methods it is indeed explained what characteristics are important for the model. However, 1) it is not explicitly mentioned which

variables are used as evaluation metrics. (2) The methods are only explained after the results are shown in which the variables are used. (But maybe this is the results of the template of the paper).

Summarized review:

In general, I believe this is a good paper with interesting work. However, it is hard to get a good overview of the results and how this work can be used in other scenarios without knowing the model used in the paper and its relation to other models.

Positive aspects:

- The paper presents a use case which really makes clear what the practical benefit of the presented models are comparing to the analytical ones. This use case shows that, because of the backpropagation, a pathological sound signal can be resorted. Which can be useful in hearing-impaired cochlea.
- The description of the architecture is elaborate. Furthermore, it is shown how different parameters of the model influence the output. Making it understandable why there is chosen for the parameters in the case of auditory model. Furthermore, the addition of the RMSE in the supplementary figures makes it easier to understand the choice for the hyperparameters to the CNN, also for readers not familiar with the auditory model.
- It explicitly discusses the limitation of having to retrain/ redesign the model when different neuron models are used.
- The availability of the data and code (which is documented) makes it easier to reproduce the results and use the framework for future work.

Suggestions for improvement:

- Although it is discussed how faster models of the auditory models can be useful in practice, it is not completely clear how the CNN can be used in combination with a cochlear implant. As the results show, in comparison to the analytical model, speedup for the IHC & ANF model is only achieved with either less frequency channels (not discussed how many frequency channels are required in a cochlear implant) or with the use of a GPU (which does not fit into an implant).
- Although a clear description is given how the parameters of the CNN are chosen in certain cases, it is hard to get to complete evaluation: 1) It is not explicitly mentioned which evaluation metrics are used. 2) In line 126-128 it is said: "Table 1 shows the final layouts of all the CoNNear modules we obtained after taking into account: (i) the L1-loss on the training speech material (i.e., the absolute difference between simulated CNN and analytical responses), (ii) the desired auditory processing characteristics, and (iii) the computational load." However, there is not given a complete clear overview of these 3 aspects or how one influences the other.
- The evaluation of the performance can be more complete. 1) It is not discussed which specific hardware is used, only that a CPU and GPU are used. 2) The highest speedup shown is achieved in combination with models from prior work. 3) The scaling of the performance in relation to complexity/ characteristics of the neuron models are not explicitly discussed.
- In lines 37-39 it is claimed that "We show here that the resulting CNN models can accurately simulate outcomes of traditional Hodgkin-Huxley neuronal models and synaptic diffusion models, but in a differentiable and computationally-efficient manner". I have the feeling that this claim is an exaggeration, as the paper shows how the CNN can simulate IHC and ANF auditory Hodgkin-Huxley-like models, and how the choosing the parameters of the CNN are chosen, with the use of a ML framework which can also make use of GPUs. It is not shown/ indicated that this will be efficient for all/ a majority of Hodgkin-Huxley models.
- The framework seems similar to simply the use of ANNs (this includes DNNs and CNNs) on any problem. This as steps (iii) to (v) [lines 342-346] from the framework is a common methodology of how to design a ANN. Furthermore, the use cases in this paper use prior work without explaining how this work is used for the decision of the initial DNN architecture (step (iii) of the framework), consequently, leaving it vague how the reader should choose the initial design of the CNN.
- In lines 59-60 it is said that: "We describe here how the CNN model architecture and hyperparameters can be optimized for such complex neuron or synapse models". However, it is not discussed how complex the models used in this paper are. Therefore, making it hard to judge what complexity the CNNs can or cannot simulate.
- In lines 130-132 it is said that: "For each CoNNear module, we first describe the set of hyperparameters that were initially selected and kept fixed, and then the hyperparameters that were

optimised to best predict performance.” However, it is not clear what the performance mentioned refers to.

- In lines 164-174 it is discussed how the window size L is important because of the adaption time constants, however, it is not discussed why this is the case.
- It is not discussed why the accuracy achieved with the use of a 400-ms inter-stimulates interval is optimal/ is good enough for this model. It would be better if it were explained which minimal accuracy needs to be achieved or what the trade-off will be in terms of accuracy vs computation time.
- In lines 192-194 it is said that: “A much deeper architecture might be necessary to simultaneously capture the characteristic ANF onset-peak and subsequent exponentially-decaying adaptation properties of ANF firing rates to step-like stimuli, and this is demonstrated in Fig. 4c” This does not give extra insight in how much deeper/ how much more computations are required to achieve the extra accuracy.
- In lines 273-274, figure 7 is referenced, however, the results of this figure are not immediately discussed. First, it is discussed how the results are obtained. This can be confusing/ unpleasant for the reader as it is not clear what kind of results are shown and what the conclusion from those results are.
- Lines 282-283 say: “Figure 7a shows that the trained CNN model accurately simulated the steady-state responses of this detailed IHC description, as reflected by the AC/DC ratio.” However, looking in Figure 7a the CoNNear model shows a significantly different AC/DC component in comparison to the other data.
- In figure 7b it is not clear what data the green dashed line represents.
- It is said in line 415-416 that the convolutional operations make architectures more hardware-friendly. However, as hardware can be any chip, I assume this should be more parallelizable, which can make the use of certain hardware (for example GPUs) more efficient, in comparison to CPUs.

Concluding remarks:

The use of a convolutional encoder-decoder architecture for simulations of Hodgkin-Huxley based auditory neural systems is interesting. This work is interesting because there is a need for faster simulations of neural models. Furthermore, the methodology for the decision of which parameters to choose for the CNN is well explained for the examples discussed in this paper. However, the work claims to be a general framework. To be able to claim generality, with high performance, the following points should be improved:

- More explicit clarification of how to use the framework. With a more explicit clarification of how the framework should be used the reader will have a better overview of how to use a CNN for neural simulations. Currently, the use of the framework is explained with the use of examples, without clearly describing how the framework should be used in other situations. Additionally, it is hard to get a complete overview of the currently used evaluation metrics. Consequently, the framework is similar to the use of ANNs/DNNs/CNNs for any problem.
- A more elaborate evaluation of the performance, in combination with accuracy. An important *selling point* is the performance of using a CNN network in comparison to the analytical model. However, the discussion misses an elaborate evaluation of the performance scaling for different problem sizes, characteristics of neural models, required accuracy (can a lower accuracy be chosen with significant faster execution times). Without an elaborate performance evaluation, the reader has a hard time to judge/ get an indication how well the model will perform on other models.

The problem I still have with this paper (after a number of good modifications by the authors) is that it presents an excellent use case and method for SNN-to-CNN conversion, and then goes on to generalize the method followed to a so-called “general framework”. However, the information provided in the paper does not suffice to support the added value (in generalizability) of the method followed, nor does it provide more specific insights on CNN design than the standard approach (choose metrics, choose network, fit, evaluate, repeat).

My suggestion: Perhaps part of the challenge the authors face is the limited space in which to present both an interesting use case (auditory-system modeling) and specific details of a general framework for SNN-to-CNN conversion. If there is no way to go into full detail on both items, it might be wise for the authors to prioritize and refocus the paper on the topic most crucial to them and/or most relevant in the context of the COMMSBIO journal. My personal opinion is (being fully aware of what this entails) that this paper should not

claim a general framework but an interesting methodology (that might be followed in other cases) and an excellent use case.

In its current state, I will have to ask again for a major revision of the manuscript.

Response to review of manuscript COMMSBIO-20-3661A – Revision 2

A neural-network framework for modelling auditory sensory cells and synapses
Fotios Drakopoulos, Deepak Baby, Sarah Verhulst

Dear Reviewer,

We thank you for the swift response to our resubmitted manuscript and the very thorough and constructive feedback. We follow your critique regarding the missing overview of the elements that are crucial for successful and accurate CNN models. While we maintained the focus of the revised manuscript on the auditory system and a representative set of auditory models, we addressed this point in depth to enable an extension of our method to other neuronal systems/synapses:

(i) We empirically derived a CNN-architecture fine-tuning guideline based on the adaptation time properties of the considered neural systems/models and the CNN-architecture receptive fields, to provide concrete design and fine-tuning steps (a framework) that can be followed when focussing on updated auditory mechanistic models or on other neurons/systems with different properties.

(ii) We better motivated and clarified our approach to fine-tune the auditory neuron/synapse CoNNear models (e.g., using the six evaluation metrics) in the manuscript text. The evaluation metrics we considered together form a classic set of auditory neurophysiology experiments that have been used in labs worldwide to characterise the properties of these neurons/synapses in vivo. This may not be well-known, so we emphasise this point more clearly in the text, and have labelled the considered metrics. By evaluating the performance of the speech-trained CoNNear on these outcomes (using the unseen auditory stimuli of the respective experiments), we determined the final set of hyperparameters that yields a “physiologically-realistic” CNN model (our aim). This fine-tuning step is inherently iterative but, when starting from a good initial architecture (see point **(i)**), it is more straightforward how to select the final CNN parameters in this evaluation procedure. At a number of places in the manuscript, we also performed additional analyses to quantify this evaluation beyond visual comparisons.

(iii) We expanded the execution-time evaluation and discussion of our CoNNear models, to provide a better indication to the reader on how the performance scales for different stimulation sizes or model characteristics.

We are using a **red** font to provide our point-by-point answers to the recent reviewer comments (black), while the reviewer comments and our answers from the previous revision are given in underlined fonts (black and blue respectively). Where indicated, line and page numbers correspond to the clean version of the revised manuscript. We think that addressing the reviewer comments improved the manuscript, and we hope that you are of the same opinion after reading the revised manuscript and our rebuttal letter.

Yours Sincerely,
Fotios Drakopoulos, Deepak Baby, Sarah Verhulst

Revisited review:

A neural-network framework for modelling auditory sensory cells and synapses

Summary:

This paper presents a model called CoNNear, with the use of a CNN with a convolutional encoder-decoder architecture, for the auditory analytical model. This model consists out of 3 stages: 1) cochlear processing, 2) IHC transduction and 3) ANF firing. In this paper the IHC transduction and ANF firing are discussed (the cochlear processing module is described in other papers). The data to train the models were obtained by analytical models, resulting in one model for IHC transduction and three models for different ANF types (low, medium and high spiking). It is discussed how the network is build and how the hyperparameters are chosen. Then the accuracy is discussed based upon plots of the output of the modules. Besides the results the accuracy the computational performance was discussed which showed 70x speedup against the reference model when using a CPU and 280x speedup when using a GPU. After this, the generalizability of the framework is shown and a use case is presented. Finally, the results are discussed and concluded.

Rebuttal Comments:

1. It is hard to judge how accurate the model is. There are shown plots, which show similar behavior to the analytically obtained data. There were mentioned about six evaluation metrics, some of which are used in the plots. However, these results were not compared to other state-of-the-art models, consequently, for someone who does not know the current state of the art in auditory sensory-cell models, it is hard to judge how accurate the presented models are. Additionally, the metrics were vaguely discussed; for example: “generally followed the pure-tone excitation patterns of the reference model (a), but showed a rather noisy response 164 across CFs, especially for the higher stimulation levels.” And “This combination of non-linearities (d) compressed the responses of the CoNNear(ANfH) and CoNNear(ANfM) models even more, and negatively affected the onset responses.” These kinds of descriptions do not give the reader a good impression what characteristics the model can simulate and what its limitations are. Giving a clear overview of how well this approach scores on the evaluation metrics in comparison to other state-of-the-art models could give the reader a good idea.

To better quantify the accuracy of our IHC-ANF models, root-mean-square errors (RMSEs) were added for the IHC excitation patterns and ANF firing rates (see Supplementary Figs 3 & 4 and RMSE section in Methods). The RMSEs were computed between the reference analytical model (Verhulst2018) and each trained CoNNear architecture and were used together with the evaluation outcomes to determine the final CoNNear architectures that optimally approximate the reference model. The “Determining the CoNNear hyperparameters” section (pages 4-7) was made more clear to justify our architecture optimisation (based on the RMSE results as well), and the evaluation metrics are now explained in more detail in Methods (lines 502-586). Difference plots were also provided for Supplementary Fig. 1, so that the accuracy of the LSR ANF CoNNear model can also be visually assessed (panel f).

The additional supplementary figures make it indeed clearer why architecture optimizations are done.

A detailed comparison between our reference analytical model and other state-of-the-art analytical models is out of the scope of this paper, but we can refer to other papers that have compared characteristics of auditory models (Saremi et al. 2016; <https://doi.org/10.1121/1.4960486> or Verhulst et al. 2018; <https://doi.org/10.1016/j.heares.2017.12.018>). The main focus of the paper was to apply our methodology to derive an accurate CNN-based approximation of the Verhulst et al. IHC-ANF model (CoNNear), as we also point out in Discussion: “Here, we focussed our development on creating CNN architectures that optimally approximate the Verhulst et al. IHC-ANF model [61], and showed how these baseline architectures can be adjusted based on the specific parameters of each analytical model and the evaluation outcomes.” However, we did compare how well our CoNNear implementation juxtaposed against two other state-of-the-art IHC-ANF models when using the same evaluation metrics (Fig. 7). We performed this comparison for the analytical models and their respective CNN approximations, and added physiological reference data when available. To provide a quantitative metric for the approximation quality, we also computed RMSEs for the CNN approximations of these two IHC-ANF models (Supplementary Fig. 5).

There are indeed made comparisons between the CNN IHC & ANF models and related work. Furthermore, in Methods (which is only placed after the whole discussion of the models) characteristics of the IHC & ANF models are discussed. However, it is not mentioned explicitly what the evaluation metrics are. For example, it is confusing that there are six evaluation metrics are initially mentioned for the IHC-ANF complex but later, in the Methods section, there are 3 metrics defined for the IHC and 3 for the ANF model. Simply stating which are the 6 metrics, before the results are shown, would make it easier for the reader to get a complete overview of the work done and accuracy achieved.

We considered 6 evaluation metrics that, together, characterise key physiological properties of the whole IHC-ANF complex, i.e., 3 for the IHC stage and 3 for the ANF stage. The specific metrics stem from classical experimental neuroscience studies, with 3 metrics referring to the presynaptic IHC receptor potential (V_m), and the other 3 to post-synaptic processing as observed in spike recordings from the ANF. Even though the IHC-ANF complex can be considered as a single structure in the auditory system, mechanistic models separate its computations into an IHC model (V_m) and a diffusion model for synaptic transmission (instantaneous spike rate). The Introduction of the manuscript was extended to better introduce the evaluation metrics and to clarify this distinction between IHC and ANF processing (lines 59-67). To make it easier to follow our evaluation procedure, we numbered each metric in Methods (from 1 to 6; pages 19-21). The metrics are now introduced throughout the text together with their respective numbers, to clarify which one is considered in the evaluation of each specific CoNNear module (Figs. 2 & 3, lines 165, 175-177, Figs. 5 & 6, lines 227, 230, lines 244-247, Fig. 7, lines 304, 321). Hence, we also point the reader to the respective subsections of Methods for more details. Unfortunately, the Methods section can only be placed after the whole discussion of the models due to the template and word limitations in the other sections.

Additionally, in lines 126-128 is mentioned that: “Table 1 shows the final layouts of all the CoNNear modules we obtained after taking into account: (i) the L1-loss on the training speech material (i.e., the absolute difference between simulated CNN and analytical responses), (ii) the desired auditory processing characteristics, and (iii) the computational load.” Although indeed different characteristics are shown and discussed, the RMSE is discussed and it can be expected that bigger CNNs require more computations; yet a clear overview is not given to the reader.

Instead, it seems they are discussed separately, which again makes it hard for the reader to get a complete overview.

We elaborated on the indicated sentence (lines 132-142) to clarify how these specific aspects were considered to determine the final architectures of each module: “The L1-loss was considered during training to determine the epochs needed to train each module and to get an initial indication of the architectures that best approximate the IHC-ANF model units. The auditory processing characteristics of each trained architecture were then evaluated on the basis of the six evaluation metrics to determine which architectures provide the most biophysically-realistic description of the IHC-ANF complex. Finally, the computational complexity of the trained architectures was taken into account to select the minimum fixed hyperparameters that yield an accurate approximation of each IHC-ANF module. Our primary concern was to develop a biophysically-realistic CNN model of the IHC-ANF complex, hence computational time was not the primary goal. However, where possible, we limited the hyperparameters of the architectures to keep the number of trained parameters (and associated computational load) as low as possible without compromising on the biophysical properties.” A few other sentences were also added to provide a better overview to the reader (e.g., lines 221-223, 244-247).

2. It is hard to judge from the paper how well the model generalizes for multiple neural models. In the text is mentioned that “Our general framework for modelling sensory-cells and synapses ...” and “All the developed CoNNear architectures can easily be integrated as part of brain networks, neuro-simulators, or closed-loop systems for auditory enhancement or neuronal-network based treatments of the pathological system. Further neural network models can be developed on the basis of the present framework to compose large-scale neuronal networks and advance our understanding of the underlying mechanisms of such systems ...”. This implies that this approach works for all Hodgkin-Huxley like models. However, in the paper it is not discussed how high the accuracy is expected to be for different kinds of models and the reader is not guided to how to choose the parameters of the DNN architecture. This is strengthened by the fact that, in the section where generalizability is discussed, only auditory neurons and synapses are discussed. Moreover, it is also mentioned that “our method might be applicable to other neuronal systems that depend on nonlinear and/or coupled ODEs”, which is weaker than the two previously mentioned statements. The authors are advised to make the text clearer about what is meant with generalizability of the framework.

The Discussion section was updated and subdivided into different subsections to clarify the framework’s capabilities. Our framework was applied to the Verhulst et al., Dierich et al. and Zilany et al. IHC-ANF models, as well as to the cochlear model of Verhulst et al. (<https://doi.org/10.1038/s42256-020-00286-8>), to show how this methodology can be used to approximate different auditory models. This attests to the generalizability of our method in the way that the reader can follow the proposed generic formula (“Framework” subsection of Discussion; lines 339-349) to model different kinds of biophysical systems. To strengthen our statements, we provided an additional example by applying our method to the standard Hodgkin-Huxley model (Supplementary Figure 6): “This provides a promising outlook because it suggests that our DNN-method might be applicable to other neuronal systems that depend on nonlinear and/or coupled ODEs (e.g., see also the application of this method to cochlear mechanics [63]). To further support this claim, we provide a simple example in Supplementary Fig. 6 where our methodology was applied to a non-auditory neuron model, the standard Hodgkin-Huxley model [1].”

The added supplementary figure, showing that besides auditory systems also the standard Hodgkin-Huxley (HH) model can be simulated (at least for a certain input), indeed adds support

to the claim of the method to be generalizable. However, it is still missing clear guidance on how the network should be adjusted/ chosen. This as it was only shown that the methodology works also for the HH model, not what changes were made to the CNN. This in combination with when discussing the performance of the method on the model of Dierich et al.: “The higher number of non-linearities comprised in this analytical model (i.e., 7 conductance branches in the Hodgkin-Huxley model) might require adaptations to the CNN architecture to accommodate this, e.g., by including an additional layer or longer filter durations to yield more accurate simulations (see also Discussion and Supplementary Fig. 5a).” (line 285-288), and “Thus, the same iterative procedure can be applied to further improve the accuracy of each supplementary CNN model, e.g., to fully capture the adaptation properties of the Dierich et al. IHC model (see Generalisability) or minimize the RMSEs of the Zilany et al. ANF models (Supplementary Fig. 5).” (line 359-360). This only gives statements (make the CNN bigger) which are common practice when applying CNNs, without giving extra insights of how much bigger the CNN needs to be and how this influences both accuracy and computation time.

Furthermore, the framework basically consists of saying that training data should come from analytical models and then use the standard approach of developing a CNN (choosing the parameters of the model). The extra insights on what a good starting point is for the CNN and how complexity of a model influences the accuracy and computational work is missing. This for example shows in line 134-136: “Prior knowledge of fine-tuning a neural-network-based model of human cochlear processing [63] helped us to make initial assumptions about the needed architecture to accurately capture the computations performed by the analytical IHC model [61].” This does not give much insight how an architecture should be chosen for new models.

Your remark makes a good point. In our design, we followed an iterative approach based on prior experience with designing CNN models for the cochlea, and we agree that the manuscript should provide a clearer guidance (or “a DIY recipe”) to apply our methods to other neuronal systems or models. We clarified the description of our followed steps and provided extra guidance as to how the reader can choose a good initial CNN architecture. To this end, two new paragraphs were added in the framework section of the Discussion (lines 384-426) that explain how an initial CNN architecture can be chosen based on the characteristics of the model/neuron that needs to be approximated. First, we explain how the adaptation time course of each analytical model can be estimated, and how this property can be used to choose a CNN architecture with a receptive field that theoretically covers the required adaptation period (lines 384-393). The initial architecture selection based on the receptive field is discussed and illustrated using the estimated adaptation properties of the different auditory models we considered in this paper (lines 393-404, 405-417 and Supplementary Table 1).

Additionally, it is also mentioned how the input window and the number of filters can be selected (lines 417-426) and we provide a more concrete description of the architecture modifications that are required to better approximate the supplementary auditory models (i.e., Dierich et al., Zilany et al.; lines 396-404). We also added a Table that summarises the properties of all the reference analytical models and the resulting CNN architectures (Supplementary Table 1) and included an “Applying the framework to other analytical models” section in Methods (pages 22-23). This section describes in more detail how the aforementioned quantities were computed and how each supplementary model was approximated. Additional details on the approximation of the standard HH model (lines 762-772) are also given in that section. The parameter selection for the CoNNear IHC model was also modified to explain the procedure more explicitly (lines 148-159).

3. A critical aspect of the work in this paper is the use of a CNN with a convolutional encoder-decoder architecture. However, it is not mentioned why this architecture is chosen and not for

example RNNs. This while it is mentioned in the text that “A different approach could be the use of recurrent layers (e.g., LSTM) within the CoNNear architectures to capture the dependency to prior stimulation without requiring long context windows”. It would be interesting if a small discussion on the architecture would be added to the paper so that the reader can better judge whether this framework can be beneficial for future work and why CNNs over other models have been preferred.

Thank you for the suggestion, we added an explicit description at the start of the Methods section (lines 411-419) which motivates our choice of adopting CNNs for the CoNNear architecture: “Our choice of using CNN encoder-decoder architectures for our model was made because of their increased efficiency and parallelism compared to other DNN architectures, such as recurrent neural networks (RNNs), that require sequential processing [81]. CNN architectures only rely on convolutions to transform input to output and are able to apply the same filter functions across multiple windows of the input in parallel [18, 82]. The same convolutional operations are applied regardless of the size of the input, making these architectures much more hardware-friendly and scalable. Recurrent layers can still be used in connection to CoNNear (e.g., to capture the dependency on prior stimulation without requiring long context windows, or when approximating other systems), but this would lead to sequential systems that are less computationally-efficient and unfit for parallel computing”

The added information about why there is chosen for a CNN is fine. Although, a comparison of RNNs vs. CNNs both in performance and accuracy would certainly be of additional value, but maybe out of scope of this paper. (Besides, this added value was shown especially in the sentence “A different approach could be the use of recurrent layers (e.g., LSTM) within the CoNNear architectures to capture the dependency to prior stimulation without requiring long context windows” which was removed from the latest paper revision).

We chose to focus our study on fully convolutional models and motivated our design choice. Other architectures might not work in the same way as CNNs and could require different modifications to fit their hyperparameters to the same reference models, and thus we think that a quantitative comparison between RNNs and CNNs is out of the scope of this paper. The indicated sentence existed in the previous version of the manuscript (and in the current one) but was moved to Methods after the revision (lines 508-511).

4. The higher performance of the presented method in this paper is one of its selling points. However, it is not mentioned how well-optimized the reference optimal implementation is and in which language this implementation is made. (It is mentioned in the methods that the Keras machine learning library with TensorFlow is used for the CNNs). Both factors can have a significant influence on the performance and can make the speedup number more or less impressive. Furthermore, the scaling of the performance numbers is not discussed. The scaling will be interesting, especially given that generalizability is advertised, as that gives the reader an idea how well this approach will perform for other neural models. Consequently, the authors are advised to improve the performance section so it will be clearer how good the presented work is.

We have expanded the description in Methods that now has a dedicated “Evaluating CoNNear execution time” section (lines 609-622), where the performance evaluation of the CNN models is discussed. We accordingly adapted the “CoNNear as a real-time model for audio applications” section as well (lines 254-269). It is correct that the achieved speed-up benefit depends on the efficiency of the reference biophysical model and the DNN architecture that is adopted to approximate it, which makes it hard to generalize the performance benefit of the method for any model, especially because of the large variety of NN methods and analytical model

implementations (see also the previous comment). The platforms where the models are executed on can also have an influence on the speed-up that can be achieved, but we used general-purpose, non-optimized frameworks to offer a “worst-case” benefit. The corresponding Methods subsection further describes the languages and evaluation methods we used: “Due to their convolutional nature, our CoNNear architectures are parallelisable, making the performance results scalable, since the same operations are applied for any input length. All stages of the reference auditory model [61] were implemented in Python, except for the cochlear stage where a C implementation was used for the tridiagonal matrix solver. Thus, both IHC-ANF implementations use general-purpose frameworks which were not optimized for CPU/GPU computing, i.e., Python for the reference models and Keras-Tensorflow-Python for the CoNNear models, making the comparison between them fair. We expect additional speed-up improvements of CoNNear when using dedicated CNN platforms in the future.”

It is indeed true that it is hard to give a performance benefit for each model. That is why it would have been nice if there was explained/showed how the CNN model scales for different numbers of input/ different sizes, so the reader can have an indication what the performance will be for different models. Furthermore, it is still not explained on which CPU and GPU the measurements are done. As for the IHC and ANF models, the speedup seems to come from using the GPU, which is indeed possible due to the used method. However, as there are also HH models with analytical GPU implementations it might be that these models would not benefit from using this methodology. Currently, the reader does not gain any information due to this. Another point of interest is that the best performance gain is showed by the CoNNear_{cochlea-IHC-ANF} model, while the CoNNear_{IHC-ANF} speedup is not that high. Consequently, it looks like the most impressive speedup is achieved with previous work, which is also not explained in this paper so the reader has to guess this.

To improve the performance evaluation of our models, we have now added a new table (Supplementary Table 2) that shows the execution speed of each CoNNear module for a longer input (full speech sentence), including the previously developed CoNNear cochlear model and a lower frequency resolution (21 frequency channels). We added a paragraph in the Limitations section of the Discussion that describes the limitations of the CoNNear models in terms of scalability (lines 468-476). Although the CPU and GPU models were mentioned in the caption of Table 2, they are now also added explicitly in the respective Methods section (lines 715-716). Additionally, a paragraph was added in the limitations section of Discussion (lines 453-467) that addresses the performance difference between the previously developed cochlear model as well as the benefits of the proposed methodology. As stated there, even for computationally-efficient implementations of analytical models and for which a speed-up benefit cannot be guaranteed, our framework can still be used to create differentiable descriptions that can be used for backpropagation purposes (lines 465-467).

5. The Discussion is just a summary of the paper and does not clearly describe the limitations. Consequently, it is hard to quickly judge how good the work is and if it should be used as there is not a clear overview compared against other related work.

We agree that the limitations of our method should be more specifically mentioned, and hence we expanded the Discussion section with a “Limitations” subsection (lines 362-394). In summary, we describe limitations related to the optimisation, training and operating limits of the resulting CoNNear models. A direct comparison against other NN-approaches to simulate neurons and synapses is difficult (and perhaps not meaningful) because there is a large variety of possible NN-implementations and none of the existing approaches focus on the approximation of analytical descriptions of auditory neurons and models. As we also mention in Discussion (lines 330-333):

“While hybrid approaches have in past studies focussed on optimising analytical model descriptions to reduce their complexity and computation effort [36–38], our method takes advantage of deep-learning to develop DNN-based descriptions that can be used for backpropagation in closed-loop systems.”

The limitations of the method used in this paper are better discussed now. However, it is still a huge limitation that by just reading the paper (and having previous knowledge of the used models) it is really hard to predict how the computation time scales for different problems and what characteristics/properties of neuron models can be modeled efficiently/accurately with the use of a CNN.

We think that the additional analysis we performed, together with the manuscript modifications, address these shortcomings. By comparing the execution time of each different model for different channel numbers (1, 21 and 201), both for fixed-sized windows (Table 2) and longer inputs (Supplementary Table 2), the reader now gets a better overview of the scalability of these architectures for different applications. Specifically, Supplementary Table 2 provides more information on how the CNN architectures scale for different simulation sizes (time or frequency sizes) or for different processing characteristics (different model stages). It is difficult to predict which exact characteristics of neuron models cannot be modelled with our CNN methodology, but we now provide a clear guideline that can be followed to approximate neuron and synapse models of similar complexity and characteristics (Framework section of Discussion), based on our experience with developing such models for auditory neurons and synapses (the focus of this paper).

6. It is mentioned that the quality of the CNN models is evaluated based on difficult properties (i.e., AC/DC ratio, excitation patterns, ANF firing rate, rate-level curves and modulation synchrony). However, these properties are not explained to the reader. The reader is advised to introduce these properties to the reader as it will make the presented work easier understandable. The same statements hold for the presented variables such as N(CF), which was used without explanation.

We agree that it was not easy to follow the evaluation metrics for readers without an explicit background in auditory neuroscience. For this reason, we described the evaluation metrics better in each subsection of Methods (lines 502-586), with reference to the experimental neuroscience papers. We also attempted to present all variables in the main text, in cases where the definition was missing (e.g., lines 87-88, 98-100, 106-107). We hope that our additions can guide both experienced and inexperienced readers to understand how we objectively evaluated the CoNNear models.

The paper is easier to read regarding the understanding of variables. However, there is still room for improvement as the acronym AN (line 54) is used without introduction. Furthermore, in Methods it is indeed explained what characteristics are important for the model. However, 1) it is not explicitly mentioned which variables are used as evaluation metrics. (2) The methods are only explained after the results are shown in which the variables are used. (But maybe this is the results of the template of the paper).

Thank you for noticing, the acronym AN is now introduced in line 54. As mentioned before, the evaluation metrics are now numbered to clarify which one is used throughout the evaluation procedure (see comment 1). Due to the template and the associated word count limit, the evaluation metrics are explained in detail in the Methods section, which is normally introduced after the main text (Intro+Results+Discussion).

Summarized review:

In general, I believe this is a good paper with interesting work. However, it is hard to get a good overview of the results and how this work can be used in other scenarios without knowing the model used in the paper and its relation to other models.

Positive aspects:

- The paper presents a use case which really makes clear what the practical benefit of the presented models are comparing to the analytical ones. This use case shows that, because of the backpropagation, a pathological sound signal can be resorted. Which can be useful in hearing-impaired cochlea.
- The description of the architecture is elaborate. Furthermore, it is shown how different parameters of the model influence the output. Making it understandable why there is chosen for the parameters in the case of auditory model. Furthermore, the addition of the RMSE in the supplementary figures makes it easier to understand the choice for the hyperparameters to the CNN, also for readers not familiar with the auditory model.
- It explicitly discusses the limitation of having to retrain/ redesign the model when different neuron models are used.
- The availability of the data and code (which is documented) makes it easier to reproduce the results and use the framework for future work.

Suggestions for improvement:

- Although it is discussed how faster models of the auditory models can be useful in practice, it is not completely clear how the CNN can be used in combination with a cochlear implant. As the results show, in comparison to the analytical model, speedup for the IHC & ANF model is only achieved with either less frequency channels (not discussed how many frequency channels are required in a cochlear implant) or with the use of a GPU (which does not fit into an implant).

Models with fewer channels can be used in the design of CI algorithms (depending on the number of electrodes) where performance benefits are also evident even for CPU execution (e.g., 21-channel simulations in Supplementary Table 2). We added a sentence that provides general ideas in this direction and think that a deeper discussion of the use of these models in CI algorithm design is out of the scope of this paper: “Thus, our modular modelling approach negatively affected the computation speed of CoNNear_{IHC-ANF}, but resulted in a biophysically-correct and rather versatile model, where the number of channels can easily be adjusted depending on the application (e.g., cochlear implants with different numbers of electrodes).” As designated chips for machine-learning algorithms gradually become available in hearing-aids and CIs, such models can be integrated to provide biophysically-realistic inputs without sacrificing the GPU runtime benefits (and associated real-time capabilities).

- Although a clear description is given how the parameters of the CNN are chosen in certain cases, it is hard to get to complete evaluation: 1) It is not explicitly mentioned which evaluation metrics are used. 2) In line 126-128 it is said: “Table 1 shows the final layouts of all the CoNNear modules we obtained after taking into account: (i) the L1-loss on the training speech material (i.e., the absolute difference between simulated CNN and analytical responses), (ii) the desired

auditory processing characteristics, and (iii) the computational load.” However, there is not given a complete clear overview of these 3 aspects or how one influences the other.

1) The evaluation metrics are now introduced in a better way throughout the text (see comment 1). 2) We elaborated on the 3 aspects in lines 133-143 (see comment 1), to provide a clear overview of how they influence the hyperparameter selection of the CoNNear modules.

- The evaluation of the performance can be more complete. 1) It is not discussed which specific hardware is used, only that a CPU and GPU are used. 2) The highest speedup shown is achieved in combination with models from prior work. 3) The scaling of the performance in relation to complexity/ characteristics of the neuron models are not explicitly discussed.

See comments 2 and 4.

- In lines 37-39 it is claimed that “We show here that the resulting CNN models can accurately simulate outcomes of traditional Hodgkin-Huxley neuronal models and synaptic diffusion models, but in a differentiable and computationally-efficient manner”. I have the feeling that this claim is an exaggeration, as the paper shows how the CNN can simulate IHC and ANF auditory Hodgkin-Huxley-like models, and how the choosing the parameters of the CNN are chosen, with the use of a ML framework which can also make use of GPUs. It is not shown/ indicated that this will be efficient for all/ a majority of Hodgkin-Huxley models.

We agree that the sentence was an overstatement and we rephrased it with a focus on auditory models (lines 38-39) so that we do not exaggerate the capabilities of the presented framework. We softened a number of additional statements throughout the text (e.g., Abstract, lines 37, 67-69, 297, 298-300, 326-328, 350-351, 354-356, 493-495, 496, 725-726, Supplementary Fig. 6), and specifically tried to avoid the use of “generalisable” framework or “generalisability” and rather used the terms “application” or “extension” for the framework. We also made the title of the manuscript more specific by adapting it to: “A *convolutional* neural-network framework for modelling auditory sensory cells and synapses”. As discussed before (and in lines 465-467), we cannot guarantee that the framework always provides faster models for any given analytical neuronal/synapse model, but it can provide differentiable descriptions that can be used in use-cases such as those presented in the paper, which offers a clear benefit over the reference analytical models (Fig. 8 & lines 329-348).

- The framework seems similar to simply the use of ANNs (this includes DNNs and CNNs) on any problem. This as steps (iii) to (v) [lines 342-346] from the framework is a common methodology of how to design a ANN. Furthermore, the use cases in this paper use prior work without explaining how this work is used for the decision of the initial DNN architecture (step (iii) of the framework), consequently, leaving it vague how the reader should choose the initial design of the CNN.

We agree that the optimisation procedure of our methodology involves several iterations and model engineering approaches that are typical in auditory modelling or other computational neuroscience/machine-learning fields. However, we believe that our hybrid neuroscience and NN approach has some benefits over a purely data-driven approach where loss minimisation is often the only decision parameter. Instead, we determine and optimise our CNN architectures based on evaluated neuroscientific (nonlinear, adaptation, level-dependent) properties of the biophysical systems. We think that such an interplay between physiological neuroscience evaluation and NN architecture optimisation is not frequently taken in the machine-learning field, and can lead to biophysically-realistic DNN-based neuronal models that can be used for backpropagation

purposes to accelerate the computational (auditory) neuroscience field. At the same time, knowledge of both the biophysical system *and* the used DNN architectures (CNN encoder-decoders) is crucial to provide insight into how this procedure can be performed in a principled way. The two newly-added paragraphs in the Discussion (lines 384-426) shed more light into how the reader can select the initial CNN architecture and perform this procedure with better-informed decisions (see also comment 2).

- In lines 59-60 it is said that: “We describe here how the CNN model architecture and hyperparameters can be optimized for such complex neuron or synapse models”. However, it is not discussed how complex the models used in this paper are. Therefore, making it hard to judge what complexity the CNNs can or cannot simulate.

We agree that the use of “complex” in the indicated sentence was unnecessary, thus we removed it from the sentence. To our knowledge and experience, the complexity of an analytical model is a rather complicated trait to quantify (apart from evaluating its execution time). However, our added discussion on how the characteristics of analytical models can be linked to the hyperparameters of the CNN architectures (lines 384-426), as well as the added performance evaluation of the execution time of each CoNNear module (Supplementary Table 2), provide some insight in this direction (see also comments 2 and 5).

- In lines 130-132 it is said that: “For each CoNNear module, we first describe the set of hyperparameters that were initially selected and kept fixed, and then the hyperparameters that were optimised to best predict performance.” However, it is not clear what the performance mentioned refers to.

Thank you for the suggestion, we changed the sentence to clarify this point and avoid ambiguity (lines 144-146): “For each CoNNear module, we first describe the initial set of hyperparameters that we kept fixed, and then motivate how the remaining hyperparameters were chosen to best predict the biophysical response properties of the reference mechanistic models.”

- In lines 164-174 it is discussed how the window size L is important because of the adaption time constants, however, it is not discussed why this is the case.

We adapted the ANF “fixed parameters” section (lines 183-223) to better present the role of each hyperparameter and the window size L , e.g.: “Since CNNs treat each input window independently, the choice of the window size L is important as it will determine the time dependencies that our ANF models will be able to encode and capture after training.” The aforementioned added paragraphs in the Framework section of Discussion also clarify the role of each ANF hyperparameter.

- It is not discussed why the accuracy achieved with the use of a 400-ms inter-stimulates interval is optimal/ is good enough for this model. It would be better if it were explained which minimal accuracy needs to be achieved or what the trade-off will be in terms of accuracy vs computation time.

Figure 4b shows the accuracy the CNN model can achieve in terms of inter-stimulus-intervals based on the selected hyperparameter values (dashed lines), and this is also explained in lines 205-207: “For a 400-ms inter-stimulus interval, the onset-peak of the HSR, MSR and LSR fibers has recovered to the 92.4%, 94.2% and 95.8% of the onset-peak of the 1.9-sec interval tone respectively (Fig. 4b).” The Limitations section of Discussion was also modified to better explain the trade-off between accuracy and computation time of the ANF model (lines 445-448):

“Depending on their application, the CoNNear ANF architectures could be extended to train using longer context or input window lengths, but this choice could sacrifice the speed-up benefits of the models while only improving the accuracy by ~4% for sustained stimulation of >400 ms or for >400 ms inter-stimulus intervals.”

- In lines 192-194 it is said that: “A much deeper architecture might be necessary to simultaneously capture the characteristic ANF onset-peak and subsequent exponentially-decaying adaptation properties of ANF firing rates to step-like stimuli, and this is demonstrated in Fig. 4c” This does not give extra insight in how much deeper/ how much more computations are required to achieve the extra accuracy.

Figure 4c shows how a 16-layer model failed to capture the adaptation properties of the ANF response, and that a deeper architecture of 28 layers was chosen to be able to capture this property. In the sentence, the “deeper architecture” refers to the 28-layer architecture that we eventually opted for. We agree that the last paragraph of the ANF “fixed parameters” section was not clear enough, so we rewrote it to avoid misconceptions for the reader (lines 209-223; see also previous comment). Additionally, the two paragraphs in the Framework Discussion section now provide additional insight into how we selected the number of layers (e.g., 393-396).

- In lines 273-274, figure 7 is referenced, however, the results of this figure are not immediately discussed. First, it is discussed how the results are obtained. This can be confusing/ unpleasant for the reader as it is not clear what kind of results are shown and what the conclusion from those results are.

Thank you for the suggestion, we adapted the whole section (Extension; lines 297-328) so that it reads more easily. The results are shown and discussed in this section and the implementation details are now moved to the new “Applying the framework to other analytical models” subsection of Methods (lines 725-744).

- Lines 282-283 say: “Figure 7a shows that the trained CNN model accurately simulated the steady-state responses of this detailed IHC description, as reflected by the AC/DC ratio.” However, looking in Figure 7a the CoNNear model shows a significantly different AC/DC component in comparison to the other data.

In this sentence, the “trained CNN model” referred to the Dierich et al. CNN approximation, which is compared to the results of the original Dierich et al. analytical model. To make this more clear, the corresponding annotations of the models are now added in the sentence, as used in the figure (Dierich2020-CNN_{IHC} and Dierich2020-IHC; lines 302-304). The same addition was done for the results of panels c and d as well (lines 322-324).

- In figure 7b it is not clear what data the green dashed line represents.

We are unsure which green dashed line you refer to. The dotted lines in Figure 7b represent the grid lines of the axes. Furthermore, the newly added orange lines represent the envelope of the response time course and the peak-to-steady-state duration (see also Supplementary Table 1).

- It is said in line 415-416 that the convolutional operations make architectures more hardware-friendly. However, as hardware can be any chip, I assume this should be more parallelizable, which can make the use of certain hardware (for example GPUs) more efficient, in comparison to CPUs.

Thank you for the suggestion, we changed the “hardware-friendly” term to “parallelisable” (lines 507-508): “The same convolutional operations are applied regardless of the size of the input, making these architectures parallelisable and scalable.”

Concluding remarks:

The use of a convolutional encoder-decoder architecture for simulations of Hodgkin-Huxley based auditory neural systems is interesting. This work is interesting because there is a need for faster simulations of neural models. Furthermore, the methodology for the decision of which parameters to choose for the CNN is well explained for the examples discussed in this paper. However, the work claims to be a general framework. To be able to claim generality, with high performance, the following points should be improved:

- More explicit clarification of how to use the framework. With a more explicit clarification of how the framework should be used the reader will have a better overview of how to use a CNN for neural simulations. Currently, the use of the framework is explained with the use of examples, without clearly describing how the framework should be used in other situations. Additionally, it is hard to get a complete overview of the currently used evaluation metrics. Consequently, the framework is similar to the use of ANNs/DNNs/CNNs for any problem.
- A more elaborate evaluation of the performance, in combination with accuracy. An important selling point is the performance of using a CNN network in comparison to the analytical model. However, the discussion misses an elaborate evaluation of the performance scaling for different problem sizes, characteristics of neural models, required accuracy (can a lower accuracy be chosen with significant faster execution times). Without an elaborate performance evaluation, the reader has a hard time to judge/ get an indication how well the model will perform on other models.

The problem I still have with this paper (after a number of good modifications by the authors) is that it presents an excellent use case and method for SNN-to-CNN conversion, and then goes on to generalize the method followed to a so-called “general framework”. However, the information provided in the paper does not suffice to support the added value (in generalizability) of the method followed, nor does it provide more specific insights on CNN design than the standard approach (choose metrics, choose network, fit, evaluate, repeat).

My suggestion: Perhaps part of the challenge the authors face is the limited space in which to present both an interesting use case (auditory-system modeling) and specific details of a general framework for SNN-to-CNN conversion. If there is no way to go into full detail on both items, it might be wise for the authors to prioritize and refocus the paper on the topic most crucial to them and/or most relevant in the context of the COMMSBIO journal. My personal opinion is (being fully aware of what this entails) that this paper should not claim a general framework but an interesting methodology (that might be followed in other cases) and an excellent use case.

In its current state, I will have to ask again for a major revision of the manuscript.

We took your concerns seriously and appreciate your viewpoints. We think that the additional simulations and analyses we performed in relating the properties of the reference analytical models more directly to good design choices for an initial CNN model architecture improved the manuscript. In our opinion, the manuscript now lives up more to the “framework” aspect by providing a more concrete and tangible method. At the same time, we maintained the focus of our

framework on modelling *auditory* sensory cells and synapses using CNN architectures. We also adjusted the manuscript title to specifically match this description: “A *convolutional* neural-network framework for modelling auditory sensory cells and synapses”.

We focussed on a set of existing auditory model descriptions that, together, resulted from decades of auditory neuroscience research. Specifically, the considered IHC membrane potential models all find their basis in classical recordings made by Palmer and Russel (1986) and Kros and Crawford (1990) and were updated over the years with data from newer experimental findings. The considered ANF synapse models find their basis in the classical model descriptions of Meddis (1986) and Westerman and Smith (1984). We think that using *framework* in this context is acceptable, because we offer CNN solutions to a number of state-of-the-art models with various levels of complexity which are well-known and representative of the auditory neuroscience field. We also strengthened the *framework* aspect by more explicitly presenting a concrete method to design CNN models for unknown or updated neuronal models, based on our presented CNN model developments for auditory neurons/synapses. At the same time, we softened our statements about the generalisability of the method to non-auditory neuronal systems (e.g., title, abstract, lines 37, 67-69, 297, 298-300, 326-328, 350-351, 354-356, 493-495, 496, 725-726, Supplementary Fig. 6), to avoid overstating the capabilities of the presented framework and keep the focus of this paper on auditory models. We also extended the performance evaluation and discussion of our CoNNear models, to provide a better overview of the application range of the methodology. We hope you agree that the updated manuscript has a much clearer focus now.

Reviewers' Comments:

Reviewer #2:

Remarks to the Author:

The points of the rebuttal have been processed well. This results in a mostly clear and well-written manuscript. Consequently, the reader can understand how this framework can be applied to other models. Furthermore, the reader gets an indication of the performance and its bottlenecks and limitations of this framework.

However, the following points could still be improved:

- A new issue is shown in lines 294-295: "Lower speed-up benefits were observed when simulating population responses for longer inputs using the single-channel IHC-ANF models". The reason why for longer inputs less speed-up benefits are achieved is not discussed.

- It is hard to draw conclusions from the Supplementary Table 1. There is no clear connection between the peak-to-steady time, RF and the number of layers, length of the filter and number of parameters. The relevant text gives the first indication. However, it is mentioned that this could be the case (lines 413-415). And maybe this is also done by the line: "A better understanding of the relative importance of units in a CNN architecture, paired with an accurate procedure to estimate the effective RF of an architecture (lines 80, 81), will yield even better-informed decisions of the initial hyperparameter selection." (lines 415-417). However, it is self-explanatory that a better understanding yields better-informed discussions and, therefore, lines 415-417 do not add much extra information. In conclusion, the whole part is somewhat confusing. It is advised to include a small extra elaboration mentioning why this is the case if a reference to the table is made.

- Lines 418-419: "as long as it contains sufficient information to accurately describe the model response.". The claim of sufficient information is vague. Being more precise would give the reader a better indication, when using this framework, if the accuracy of the results is due to the size of the input window or other parameters of the CNN.

- "Architecture" is spelled wrong in line 756.

- In figure 7c and 7d, two green lines are shown. However, there is only one green line in the legend.

The first issue has to be addressed. The remainder are minor issues. Therefore, my conclusion is that the paper should be accepted and the authors should agree to address the points mentioned above.

Response to review of manuscript COMMSBIO-20-3661B – Final Revision

A convolutional neural-network framework for modelling auditory sensory cells and synapses

Fotios Drakopoulos, Deepak Baby, Sarah Verhulst

Dear Reviewer,

We thank you once again for the swift response to our revised manuscript and the constructive feedback. We are using a blue font to provide our point-by-point answers to the reviewer comments (black).

Yours Sincerely,
Fotios Drakopoulos, Deepak Baby, Sarah Verhulst

Reviewer #2 (Remarks to the Author):

The points of the rebuttal have been processed well. This results in a mostly clear and well-written manuscript. Consequently, the reader can understand how this framework can be applied to other models. Furthermore, the reader gets an indication of the performance and its bottlenecks and limitations of this framework.

However, the following points could still be improved:

- A new issue is shown in lines 294-295: "Lower speed-up benefits were observed when simulating population responses for longer inputs using the single-channel IHC-ANF models". The reason why for longer inputs less speed-up benefits are achieved is not discussed.

We agree that the newly-presented results could be described better. Although some aspects were briefly discussed in the caption of the Supplementary Table 1 and the Discussion, we further added a few explanatory sentences in the "CoNNear as a real-time model for audio applications" section (lines 291-298): "Similar speed-up benefits were observed when simulating single-unit responses for longer inputs on the CPU (Supplementary Table 1), with the ANF models providing a somewhat faster execution. The achieved speed up was more significant on the GPU, with a 1600 times faster CoNNearIHC execution than the reference IHC model and a ~ 550 times faster CoNNearANF model execution on average. Different from Table 2, the single-channel CoNNearIHC-ANF models were used for all simulations of longer inputs to avoid large memory allocation. This resulted in lower speed-up benefits for the population-response simulations of Supplementary Table 1, but, for high-end systems that can support this additional memory requirement, the parallel simulation of all N_{CF} channels can provide a speed-up benefit comparable to the results of Table 2."

- It is hard to draw conclusions from the Supplementary Table 1. There is no clear connection between the peak-to-steady time, RF and the number of layers, length of the filter and number of parameters. The relevant text gives the first indication. However, it is mentioned that this could be the case (lines 413-415). And maybe this is also done by the line: "A better understanding of the relative importance of units in a CNN architecture, paired with an accurate procedure to estimate the effective RF of an architecture (lines 80, 81), will yield even better-informed decisions of the initial hyperparameter selection." (lines 415-417). However, it is self-explanatory that a better understanding yields better-informed discussions and, therefore, lines 415-417 do not add much extra information. In conclusion, the whole part is somewhat confusing. It is advised to include a small extra elaboration mentioning why this is the case if a reference to the table is made.

To make this paragraph more clear, the indicated sentence was moved to the Methods (lines 754-756) and some explanatory text was added in the caption of Supplementary Table 2 to better describe the presented parameters: "For each CNN architecture, the RF lengths were computed from the selected N_{enc} and filter length parameters using Eq. 4 and were converted to ms durations ($f_s = 20$ kHz for all CNN models). Choosing architectures with longer RFs than the computed peak-to-steady-state durations resulted in models that accurately approximated the respective analytical descriptions. The filters per layer and total number of trainable parameters of each CNN model are also shown for comparison purposes."

- Lines 418-419: "as long as it contains sufficient information to accurately describe the model response.". The claim of sufficient information is vague. Being more precise would give the reader a better indication, when using this framework, if the accuracy of the results is due to the size of the input window or other parameters of the CNN.

We agree that the indicated sentence was not clear, thus we modified it (lines 418-420): "Finally, depending on the time course of the response of an analytical model and the selected number of encoder layers N , any multiple of 2^N samples can be chosen as input window for the training, as long as it captures the full response time course of the analytical model for a broad range of stimulation paradigms.

- "Architecture" is spelled wrong in line 756.

Thank you for noticing, we corrected this.

- In figure 7c and 7d, two green lines are shown. However, there is only one green line in the legend.

The two line-types (dashed and solid) for each label in Figures 6 and 7 correspond to the results for tonal stimuli of 1 and 4 kHz respectively, as indicated in the captions of the two figures.

The first issue has to be addressed. The remainder are minor issues. Therefore, my conclusion is that the paper should be accepted and the authors should agree to address the points mentioned above.